


# Spring melting season methane emissions in northern high latitude wetlands are governed by the length of the season and presence of permafrost

Sara Hyvärinen[1], Maria K. Tenkanen[1], Aki Tsuruta[1], Anttoni Erkkilä[1], Kimmo Rautiainen[1], Hermanni Aaltonen[1], Motoki Sasakawa[2], and Tuula Aalto[1]

[1]Finnish Meteorological Institute, 00101 Helsinki, Finland
[2]National Institute for Environmental Studies, Ibaraki, Japan

**Correspondence:** Sara Hyvärinen (sara.hyvarinen@fmi.fi)

**Abstract.** Northern high latitude wetlands are significant sources of methane, with emissions driven by seasonal soil freezing and thawing. To better understand the seasonality of northern high latitude methane emissions, we defined the spring melting season using the remote sensing Soil Moisture and Ocean Salinity Freeze/Thaw data from 2011–2021. To estimate methane emissions in the northern high latitudes, we used the atmospheric inverse model CarbonTracker Europe-$CH_4$. The melting season was defined for three permafrost zones and for seasonally frozen non-permafrost region using two approaches: region-based, which considered climatological conditions of permafrost regions, and grid-based, which defines the melting season at a finer $1° \times 1°$ scale.

The length and timing of the melting season varied significantly depending on the approach. The melting season generally occurred between March and June and was influenced by the air temperature, with a negative correlation between the length and the mean temperature. The longest melting season was in the non-permafrost zone and the shortest varied between the two methods. The spring melting season emissions were on average 1.83 Tg with the region-based approach and 0.45 Tg with the grid-based approach, the non-permafrost zone having the largest share of the spring emissions. The emissions were largely dependent on the season's length. Year-to-year variation was modest, within 15% (region-based) and 23% (grid-based) of average emissions, and there was also no trend during the study period. Our dual-method approach allows for robust comparison with both large-scale regional studies and localized site-level research.

## 1  Introduction

Northern high latitude wetlands are an important and dynamic part of the climate system (Hugelius et al., 2020). They are a large source of methane ($CH_4$), which is the second most important anthropogenic greenhouse gas causing climate change after carbon dioxide ($CO_2$), having a 29.8 times stronger global warming potential in 100-year timescale than $CO_2$ without considering climate feedbacks (Forster et al., 2021). A large portion of the total soil carbon is stored in northern wetlands and the underlying permafrost (Scharlemann et al., 2014; Hugelius et al., 2020) but due to climate change and Arctic amplification, thawing permafrost could affect the carbon stock (Schuur et al., 2015; Knoblauch et al., 2018; Voigt et al., 2019; Turetsky



et al., 2020). Even if the permafrost does not fully thaw, the deepening of the active layer – the top layer of soil that thaws in summer and freezes in winter – can still release a significant amount of carbon. Permafrost thaw will likely lead to soil drying and increased drainage, potentially accelerating organic matter decomposition and $CO_2$ emissions while suppressing $CH_4$ emissions (Lawrence et al., 2015) but the effect of the released $CH_4$ might be as large due to its stronger global warming potential (Schuur and Abbott, 2011). According to Neumann et al. (2019), increasing rainfall and warming soils could increase near-term global warming, and the total annual boreal methane emissions could rise 4 Tg per year. The study by Poulter et al. (2017) using biochemical model concluded that boreal wetland $CH_4$ emissions have already increased by 1.2 Tg yr$^{-1}$ between 2000–2012.

In addition to permafrost thaw, Arctic amplification is expected to significantly impact snowmelt and seasonal soil thawing, including the active layer, in northern high latitude wetlands. There are still a lot of open questions related to the future of methane emissions in the northern high latitudes due to uncertainties in the wetness of the area and possible feedback loops. For instance, de Vrese et al. (2023) demonstrated that atmospheric feedbacks resulting from increasingly dry Arctic may offset the effects of growing wetland extent and result in comparable $CH_4$ emissions as in the case where the Arctic would remain wet. The changing Arctic hydrology affects the spring melting season emissions as well. The hydrological conditions during the spring melting season in the northern high latitudes are especially uncertain due to the unreliability in the amount of snow as well as the melting and evaporation of snow. This lack of certainty brings unpredictability to the melting season methane emissions.

$CH_4$ emissions from the northern high latitude wetlands are low during winter when the soil is frozen and high during summer after the soil has thawed (Aselmann and Crutzen, 1989; Rinne et al., 2018), also depending on the soil temperature and hydrological factors (Rychlik, 2009; Zhu et al., 2013; Turetsky et al., 2008). During the spring thawing, the emissions increase rapidly. Typically, the $CH_4$ storage is low during the spring thawing after the dormant cold season, and the $CH_4$ is released gradually as the soil melts (Raz-Yaseef et al., 2017). However, previous studies have reported large bursts of $CH_4$ from the wetlands during the spring thawing (Jin et al., 1999; Tokida et al., 2007; Song et al., 2012; Raz-Yaseef et al., 2017), though the reports are scarce with only a few measurement sites and just a few years of flux measurements from before the thawing starts (e.g. winter flux data) (Raz-Yaseef et al., 2017). The springtime bursts of $CH_4$ have been linked to rain-on-snow events which enhance soil cracking (Raz-Yaseef et al., 2017). However, the frequency and impact of these pulses to spring methane emissions are still highly uncertain. Our research did not focus on these springtime bursts, but that does not negate their existence. We focused on large scale methane emissions in the northern high latitudes.

Previous studies have mostly focused on methane emissions during the growing season while the other seasons, such as the autumn freezing period, winter season and the spring thaw season, has received less attention (e.g. Vourlitis et al., 1993; Sachs et al., 2008; Zona et al., 2009; Parmentier et al., 2011). The regional studies focusing on the spring season, have often defined the timing of the spring thaw season simply as certain months, e.g. March, April and May (e.g. Castro-Morales et al., 2018; Ito et al., 2023) or in a case of site-level studies, using site-specific soil temperature measurements (e.g. Zona et al., 2016; Bao et al., 2020; Tagesson et al., 2012; Raz-Yaseef et al., 2017). Using soil temperature to define the soil thawing would be ideal, but a reliable soil temperature data is available only from point-wise measurements. Thus, to be able to study spring season





on a regional scale, we need to use a proxy for the soil thawing. In this study, we define the spring melting season using the remote-sensed Soil Moisture and Ocean Salinity (SMOS) soil Freeze/Thaw (F/T) data (Rautiainen et al., 2016). The SMOS F/T data provides daily information of the freezing and thawing state of the soil in the northern high latitudes at the resolution of 25 km. Using the SMOS F/T data gives us a dynamic picture of the soil thawing including the active layer of the permafrost during the spring and enables us to define the melting season for the whole northern high latitude regions instead of using a static definition (e.g. specific months). The SMOS F/T data have been used successfully to define summer thaw, autumn freezing, and winter cold seasons in the northern high latitude wetlands (Tenkanen et al., 2021; Erkkilä et al., 2023). In this paper, the SMOS F/T data is used for the first time to define the spring melting season.

In addition to in-situ measurement based studies, the spring $CH_4$ emissions has been studied with process-based models (Ito et al., 2023; Castro-Morales et al., 2018). Process-based models estimate methane fluxes by simulating physical, chemical, and biological processes. Another type of modeling is inverse modeling. Inverse models are statistical approaches which can be used to inform and "re-evaluate" the process-based estimates and decrease the uncertainties in $CH_4$ emissions using atmospheric $CH_4$ measurements, so that the differences between simulations and measurements is minimized (Wittig, 2023).

The aim of this study is to estimate the spring melting season and its $CH_4$ emissions in the northern high latitude permafrost and wetland regions. We use the SMOS F/T data to define the spring melting season and determine methane emissions with the global atmospheric inverse model Carbon Tracker Europe-$CH_4$ (CTE-$CH_4$) (Tsuruta et al., 2017) at high spatial resolution of $1°$ latitude $\times$ $1°$ longitude. The years studied are 2011–2021 due to availability of the SMOS data. Previously, CTE-$CH_4$ has been used to study northern high latitude wetland emissions for various seasons (Tenkanen et al., 2021; Erkkilä et al., 2023), but this is the first time with a focus on spring melting season. The spring melting season is defined for four different permafrost zones (sporadic, discontinuous and continuous permafrost and a seasonally frozen zone non-permafrost), as well as individually for each $1° \times 1°$ grid cell to determine the melting season emissions both in climatological and local scales.

## 2 Materials and methods

### 2.1 SMOS F/T soil state estimates

To define the melting season of the northern high latitude wetlands, we used the European Space Agency's (ESA) SMOS Soil Freeze and Thaw State product (version 3.0) (Rautiainen et al., 2016; European Space Agency, 2023). It provides daily information on the soil state in the northern latitudes based on observations from the ESA SMOS satellite (Kerr et al., 2010). The SMOS payload instrument Mircowave Imaging Radiometer with Aperture Synthesis measures the natural microwave radiation emitted by the Earth's surface at L–band (f $\approx$ 1.4 GHz) (Kerr et al., 2010). The intensity of the natural microwave radiation is related to the physical temperature of the target and its emissivity; it is quantified by the brightness temperature. Detection of the soil F/T state with SMOS data is based on the high contrast in permittivity between free liquid water and ice at L–band (Rautiainen et al., 2016). Soil containing liquid water has a higher permittivity and therefore lower emissivity than frozen soil. This causes the brightness temperature of thawed soil to be lower than that of frozen soil. L–band measurements are especially useful for soil observations due to their relatively long wavelength because it is less affected by vegetation



cover and cloud layers compared to higher frequencies. Additionally, the deeper penetration depth of L–band microwaves enables observations not only from the surface skin but also from the top 0–5 cm soil layer. Importantly, passive microwave observations are not dependent on sunlight, which is important for determining the melting period in the northern high latitudes since the melting season can start already during the time when sunlight is limited due to the polar night.

The SMOS F/T soil state detection algorithm utilizes Centre Aval de Traitement des Données SMOS daily gridded level 3 brightness temperature data (Al Bitar et al., 2017) as input data. Additionally, two ancillary data sets were used: European Centre for Medium–Range Weather Forecast (ECMWF) ERA5 2 m air temperature (Hersbach et al., 2020a) and National Snow and Ice Data Center Interactive Multisensor Snow and Ice Mapping System snow cover (U.S. National Ice Center, 2004). The F/T detection algorithm is threshold-based: observations are compared to empirically defined frozen and thawed soil references

and categorized as frozen, partially frozen, or thawed soil based on these thresholds. Of the three categories, the thawing state of the soil is used in this study to define the melting season. Two distinct soil state products are provided: one using ascending orbit observations (6 AM) and the other using descending orbit observations (6 PM). In this study, we used only the ascending orbit data, as they are less affected by radio frequency interference (RFI) over the Eurasian continent and were better for the purpose of our study since the melting snow during daytime affect the descending orbit observations in the evening. The SMOS

F/T data are available online from the ESA SMOS dissemination server (ESA) and from FMI (SMOS). The data are available from July 2010 until the current day with a latency of a couple of days. Its resolution is 25 km × 25 km. To combine with inverse model results, the resolution of the data was changed to $1° × 1°$ resolution by selecting the pixels whose centre was inside the $1° × 1°$ grid cell and calculating the fraction of pixels that were defined as thaw.

## 2.2 CarbonTracker Europe-CH$_4$

The methane fluxes were estimated with the CarbonTracker Europe-CH$_4$ (CTE-CH$_4$) inverse model (Tsuruta et al., 2017). In its assimilation phase, a Bayesian cost function was minimized:

$$J = (\mathbf{x} - \mathbf{x}^b)^T \mathbf{P}^{-1} (\mathbf{x} - \mathbf{x}^b) + (\mathbf{y} - H(\mathbf{x}))^T \mathbf{R}^{-1} (\mathbf{y} - H(\mathbf{x})), \tag{1}$$

where $\mathbf{x}$ is a state vector which contains a set of scaling factors that multiply the prior CH$_4$ surface fluxes, which are meant to be optimized, and $\mathbf{x}^b$ is the prior state vector. $\mathbf{P}$ is the state vector error covariance matrix. $\mathbf{y}$ is a vector consisting of the

atmospheric methane observations and $\mathbf{R}$ is the error covariance matrix of the observations $\mathbf{y}$. $H$ is an observation operator, which is an atmospheric transport model TM5 in this study (see Section 2.2.1).

The model used the ensemble Kalman filter (EnKF) (Evensen, 2003; Peters et al., 2005) data assimilation scheme within the CarbonTracker Data Assimilation Shell (CTDAS) (van der Laan-Luijkx et al., 2017) with an ensemble size of 500 and a 5-week lag to optimize the fluxes (Peters et al., 2005; Tsuruta et al., 2017). The anthropogenic and natural fluxes were optimized

separately but simultaneously in a weekly temporal resolution. The fluxes in the high northern latitudes were optimized at the spatial resolution of $1° × 1°$, and regionally elsewhere. The spatial correlation followed an exponential decay model (Peters et al., 2005) with correlation lengths of 100 km for $1° × 1°$ grid-based domains, 500 km for other land domains, and 900 km for oceanic domains. Anthropogenic and natural CH$_4$ fluxes were assumed to be uncorrelated, as were land and ocean domains.



### 2.2.1 TM5 chemistry model

TM5 is an atmospheric chemistry transport model (Krol et al., 2005). Here, it was used as the observation operator to transform the methane fluxes to atmospheric mole fractions. In this study, its global horizontal resolution was $4° \times 6°$ with an intermediate zoom region of $2° \times 3°$ (14° N–82° N, 36° W–54° E) framing a $1° \times 1°$ zoom over Europe (24° N–74° N, 21° W–45° E). The model used preprocessed meteorological data from ECMWF ERA5 reanalysis data with a 3-hour resolution (Hersbach et al., 2020a). The vertical domain was divided into 25 hybrid sigma pressure levels from the surface to the upper atmosphere. The

chemical loss of $CH_4$ in the atmosphere to the sinks of OH, was based on monthly precalculated values by Houweling et al. (2014), and Cl and $O(^1D)$ sinks were based on the atmospheric chemistry general circulation model ECHAM5/MESSy1 (Jöckel et al., 2006; Kangasaho et al., 2022). The variability of the atmospheric sinks between different years was not considered, but varied monthly, and the sinks were not optimized.

### 2.2.2 Observations

In addition to the observations from the ObsPack v4.0 (Schuldt et al., 2021; Masarie et al., 2014), observations from two stations in Finland (Kumpula, Sodankylä) (Tsuruta et al., 2019) and from nine stations in Siberia (Sasakawa et al., 2010, 2025) were used. All stations are listed in the Appendix A1 and the location can be seen in Fig A1. Globally, 183 stations had observations between 2011-2021, with some stations having two or three institutions contributing. The data included weekly discrete air samples and hourly continuous measurements, and the data was filtered according to the institutions' quality flags. Only data

points that represented well-mixed conditions were included, which means that daily averages were calculated from 12 to 4 pm local time, except for high mountain sites where averages are calculated from 0 to 4 am local time, following Tsuruta et al. (2017).

Observational uncertainties, or "model–data mismatches", were estimated for each site based on site-specific factors, measurement accuracy, and the capability of TM5 to simulate atmospheric $CH_4$ mole fractions (Bruhwiler et al., 2014; Tsuruta

et al., 2019). These discrepancies arose from TM5's resolution and transport errors, with e.g. better performance at remote marine sites compared to those affected by strong local emissions. Sites were categorized, for example, as marine boundary layer (4.5 ppb), terrestrial (25 ppb), mixed marine and terrestrial (15 ppb), and strong local influence (30 ppb). Uncertainties ranged from 4.5 to 75 ppb.

### 2.2.3 Prior fluxes

The prior anthropogenic emissions were taken from the Emission Database for Global Atmospheric Research (EDGAR v6.0) (Monforti Ferrario et al., 2021). The emissions from LPX-Bern DYPTOP v1.4 (Lienert and Joos, 2018) were used as the natural biospheric prior emissions. Methane emissions from other sources were: Weber et al. (2019) for ocean, the Global fire emission database (GFED v4.1s) (van der Werf et al., 2017; Randerson et al., 2017) for biomass burning, and VISIT (Ito and Inatomi, 2012) for termites. Other fluxes, excluding biospheric and anthropogenic fluxes, were not optimized.



For both the anthropogenic and biospheric fluxes, we used the prior uncertainty of 80% for terrestrial fluxes and 20% for oceanic fluxes, assuming uncorrelated uncertainties, following previous studies (e.g., Tsuruta et al., 2017; Bruhwiler et al., 2014).

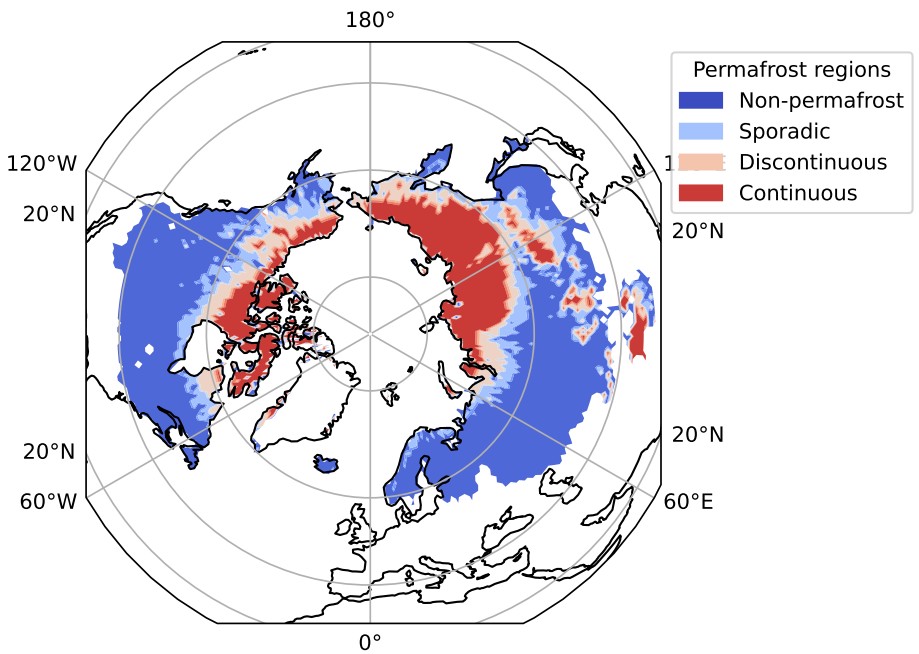

**Figure 1.** Extents of the four permafrost zones in $1° \times 1°$ resolution.

### 2.3  Permafrost map

The permafrost extent (v3.0) is one of the variables belonging to ESA's Climate Change Initiative (CCI) (Bartsch et al., 2020;
Obu et al., 2019). The temporal resolution of the data is one year and the spatial resolution is 926.63 m ((Obu et al., 2021)). Using the permafrost and SMOS F/T soil state data, we split the northern latitudes into four zones depending on how much of the exposed land surface is underlain by permafrost: 90–100% for the continuous permafrost, 50–90% for the discontinuous permafrost, 14–50% for the sporadic zone and the non-permafrost zone mainly comprising areas that were frozen seasonally based on the SMOS F/T data with some potential permafrost spots, since the CCI permafrost data had not values lower than
14% (Fig.1). Areas where no SMOS F/T data was available, were excluded from this study.

The permafrost data was only available until 2018. Average permafrost zones were calculated from the years 2011–2018. The four permafrost zones had only minor changes between the different years. The average permafrost percentage was calculated for each $1° \times 1°$ grid cell after the resolution was changed to $1° \times 1°$ from the 1 km resolution by taking mean for each grid. The average areas were used in this study to define the four permafrost zones for all the years from 2011–2021. We are aware



that a newer version of the permafrost data (v4.0) including the years 2019–2021 has been published but it was not used in this study (Westermann et al., 2024).

## 2.4   ERA5 2 m temperature data

ERA5 is the latest global reanalysis from the ECMWF available from 1959 to the present day (ECMWF; Hersbach et al., 2020b). It provides hourly estimates of multiple land, oceanic and atmospheric climate variables. Here, the ERA5 2 m air
temperature reanalysis data was used to study the relationship between the air temperature and CTE-CH$_4$ emissions during the melting season. The ERA5 temperature data was additionally used as an auxiliary dataset in the SMOS F/T data and as input data in the CTE-CH$_4$.

## 2.5   Defining the melting season and its methane emissions

### 2.5.1   Definition of the melting season

In this study, the melting season was defined as a season in the northern latitudes in spring, when the soil in a specific region turned from frozen to thawed based on the "thawed" state in the SMOS F/T data (see Section 2.1). The word "melting" is used here instead of "thawing" because the SMOS F/T data can indicate the melting of the snow instead of the soil, especially at the beginning of the melting season. This is because the microwave radiation signal from wet snow resembles the signal from thawed soil (Rautiainen et al., 2016). However, methane emissions are possible even at the very beginning of the melting
season because the air temperature rises above zero and melted water can trickle into the soil. It is thus justified to start the melting season from the melting of the snow. The boundaries used in this study were similar to the ones used by Erkkilä et al. (2023) to define different seasons in the northern high latitude wetlands.

     The melting season was defined for four permafrost zones: non-permafrost, sporadic, discontinuous and continuous permafrost (region-based approach), and separately for each 1° × 1° grid cell (grid-based approach) in these zones. The region-
based approach gives information about the permafrost areas constrained by their specific climatological conditions while the grid-based melting season was studied to illustrate the local variation in the melting of the soil.

     With the region-based approach, the melting season was set to start 1) when the mean thawing fraction of a permafrost zone had surpassed the minimum thawing fraction of that year by 0.1 (thaw(%) $\geq$ thaw$_{\text{min,year}}$(%) $+ 10\%$), and 2) the last day in the spring, when the minimum annual mean thawing fraction was reached, was surpassed. During some years, the
freezing of the soil continued past the turn of the year, which meant that the first boundary was reached before the maximum freezing of the soil. This meant that an additional condition for the beginning of the melting season had to be defined. With the second condition, the melting season could be separated from the autumn freezing period. In regions with permafrost, the first condition was surpassed later during the spring than the second condition, but in the non-permafrost zone the second condition was needed.

The season ended when the mean thawing fraction of the whole zone had surpassed 0.8 of the maximum thawing fraction of all years which was 1 for all zones (thaw(%)/thaw$_{\text{max,all}}$(%) $\geq 80\%$). The fraction of 0.8 was chosen because there was not as





much variation in the thawing fraction that late in the spring indicating a stable thaw state. The day when the melting season ended was included in the melting season. The size of the grid cells in the north-south direction was taken into account when defining the melting season. The thawing fraction in each grid cell was defined from the amount of thawed pixels in a grid cell.

The amount of pixels in a grid cell diminished towards the north. The mean thawing fraction of all grid-cells in a permafrost zone was then calculated.

In the grid-based approach, the start of the melting season was defined differently. The season started when one SMOS F/T pixel (25 km × 25 km) in the 1° × 1° grid cell had melted. There were maximum of 18 SMOS F/T pixels and minimum of 1 pixel in each 1° × 1° grid cell depending on data availability and geographic location. For example, in a grid cell with 18

pixels, the season started when the thawing fraction was 1/18, meaning that 5.6% of the area of that grid cell has melted. The season ended when the thawing fraction in the grid cell had surpassed 0.8 of the maximum thawing fraction of that year. To separate the spring melting season from the autumn freezing season, the melting season was defined to start and end before the 212th day of the year. There were grid cells where there was missing data on some days. The missing data was replaced by interpolating linearly between the previous and the coming day with an existing value. If none of the pixels melted in a

grid cell, or if the thawing fraction during the spring never became below 0.8 of the maximum thawing fraction of that year, meaning that the grid cell did not freeze, the grid cell did not have a melting season. In some of the grid-cells, the condition for the end of the melting season was never surpassed before the 212th day of the year even though the season had a beginning. For those grid-cells, the end of the melting season was defined as a day when the soil had thawed the maximum amount during the spring season. However, this only happened in 2 grid-cells in 2017. Additionally there were a couple of grid-cells where

the minimum thawing fraction during the spring time was reached after the maximum was already reached. In those grid-cells the season was defined to begin before the maximum was reached. Excluding the first three years, fewer than 1% of grid cells in the study area did not have a melting season annually. In the first three years, this percentage ranged from 6.5% in 2011 to 1.1% in 2013.

### 2.5.2   Calculating melting season methane emissions

After defining the melting season, the methane emissions were calculated from the CTE-CH$_4$ inverse model biospheric posterior emission estimates. The methane emissions were optimized weekly, and to calculate the daily emissions, the model data was interpolated linearly from one optimized weekly value to the next weekly value. From these daily values, the posterior methane emissions were analysed during the melting season. Similarly to the melting season, the regional emissions were calculated separately for the four permafrost zones: non-permafrost, sporadic, discontinuous and continuous permafrost zones.

The total melting season CH$_4$ emissions in a region or grid cell in teragrams of methane (Tg CH$_4$ per region per season, hereafter denoted simply as Tg) were calculated from the daily methane emission estimate. The daily values of all grid cells were added together in each permafrost zone during the melting season. The average emissions were calculated in the unit of μgCH$_4$ m$^{-2}$s$^{-1}$ by dividing the total emission by the area of the specific permafrost zone and the length of the melting season. The emissions were studied against the land area of the permafrost zones instead of the actual area of wetlands or permafrost



in each zone. This was done because the exact area of wetlands is uncertain (Saunois et al., 2020), and using the estimated wetland area extent would have added another source of uncertainty.

As the melting season was defined separately for each $1° \times 1°$ grid cell, the melting season emissions were calculated separately as well. The sum of the emissions per grid cell per melting season was calculated in each grid from the first day of the melting season to the last. To calculate average of the methane emissions in the unit of $\mu gCH_4 \ m^{-2}s^{-1}$, the emissions in each grid were divided by the respective area of the grid cell and the length of its melting season.

## 3 Results

### 3.1 Length of the melting season

The average length of the melting season was approximately four times longer when the region-based approach (45 days) was used than when the grid-based approach was used (10 days) (Fig. 2). Using the grid-based approach, the average length of the melting season was the longest in the southernmost zone, non-permafrost (12 days), and the shortest in the northernmost zone, continuous permafrost (7 days) (Fig. 2 and Fig. 3). For the region-based melting season, there was no as clear gradient in the north–south direction as there was in the grid-based mean lengths. However, the longest melting season was typically still in the southernmost permafrost zone, the non-permafrost (57 days), except for 2011, when the longest season was in the continuous permafrost zone (54 days), and in 2018, when it was the longest in the sporadic zone (57 days). On average, the region-based season was the shortest in the sporadic and discontinuous zones (40 days), which also had smaller areas than the other two zones. The years with the longest and shortest melting seasons differed between the two methods and grid-based and region-based lengths of the melting season also did not strongly correlate (Fig. A2).

Most of the of grid-based melting seasons lasted only a few days (Fig. 4). However, some grid cells had a very long melting season, some as long as the region-based melting season. For example, the West Coast of Canada and the USA had large areas with a longer melting season (Fig. 3). Other areas with noticeably longer melting season include the coast of Norway, Iceland, East Coast of the USA, Mongolia, the southern parts of Russia and Northern China. The shortest melting season in a grid cell was one day in all the zones and the longest season was found in the sporadic zone and it lasted 212 days, the length of the the maximum duration of the melting season based on our method. It was clearly an outlier, since 95% of grid cells in the sporadic zone had a melting season season shorter than 31 days. In addition, 95th percentiles for other zones were 40 days in the non-permafrost zone, 26 days in the discontinuous permafrost zones, and 20 days in the continuous permafrost zone.

With the region-based approach, the length of the melting season had a larger inter-annual variation (7–17 days depending on the permafrost zone) than the average length of the grid-based season (Fig. 2), which might be linked to the inter-annual variations of weather in the permafrost zones. Hence, the grid-based method might be better at detecting the actual melting of the snow and soil, because it detects the faster changes in a specific grid-cell. An example of inter-annual variation of weather affecting the region-based length of the melting season is in 2011, when in the continuous permafrost zone, the average ERA5 2 m mean air temperature was the lowest and the melting season was the longest. In contrast, in 2017 and 2019, the 2 m mean temperature was higher and melting season shorter in the continuous permafrost zone. In 2013, the spring melting season





was the shortest in the sporadic and discontinuous permafrost zones. According to the ERA5 2 m air temperature data, the melting season mean temperature was higher in these two zones than during other years of our study period. In 2015, the mean

temperature on these two zones was higher during the spring melting season as well. This suggest that the melting season was shorter and warmer after the colder weather surpassed. In 2017–2019, the sporadic zone had its longest melting season while the mean temperatures were the lowest. This all indicates that the melting season timing and length were linked to the mean temperature of the zone.

The longer melting season in the southern regions could have been caused by an early onset of the melting season, followed
by variation between negative and positive temperatures, which would have slowed down the melting. The early onset of the melting season and the consecutive variation between temperatures would have made the mean temperature lower and the melting season longer. To confirm this, the relationship between the length of the melting season and mean temperature was studied (Fig. 5). A negative correlation was found between the two variables, which was the strongest in the discontinuous permafrost zone (region-based $p = 0.008$) and sporadic zone (grid-based $p = 0.005$). With the region-based approach, the cor-
relation between the variables was significant ($p < 0.05$) for all the zones and there was a more prominent negative correlation between the length and the starting day of the melting season in all the permafrost zones ($p < 0.001$ for all the zones, Fig. A3). This indicates that the later the season started, the shorter the melting season was, at least in the larger permafrost zone scale. Additionally there was a positive correlation between the region-based starting day of the melting season and the mean temperature of the melting season (sporadic: $p < 0.001$, continuous: $p < 0.01$, and non-permafrost and discontinuous: $p < 0.05$).
This all indicates that the melting season started earlier, when the mean temperature was lower, on a larger scale.

With the grid-based approach there was not as strong correlation between the mean starting day of the melting season and the mean temperature or mean length of the melting season ($p \geq 0.1$ for most of the permafrost zones). Additionally, the melting season mean temperature was higher with the grid-based approach than with the region-based approach (Fig. 5). The mean values of the length and temperature of the grid-based melting season might not have been the best to describe the relationship
between the length of the melting season and the temperature of the melting season because the variation between different grid cells is not seen.

### 3.1.1 Start and end days of the melting season

With the region-based melting season, the start and end of the season varied from year to year in each permafrost zone (Tab. A2). The range of variation of the starting days was 21 days in the non-permafrost and sporadic zones, 15 days in the discontinuous
permafrost zone, and 18 days in the continuous permafrost zone. The range of variation of the ending days was shorter, only 6 days in the non-permafrost zone, 14 days in the sporadic zone, and 8 days in the discontinuous and continuous permafrost zones. Between different grid-cells, the start and end of the melting season varied more within a year than the averages from year to year. However, even inside one grid-cell, there was more inter-annual variation that was not seen from the average (Fig. A4). On average, the region-based melting season started earlier and ended later than the grid-based melting season. However,
in some grid-cells, the melting season started much earlier and ended later than the region-based season, as expected since at least 10% of the grid-cells had to be thawed for the region-based melting season to start and 80% had to be melted for it to end.





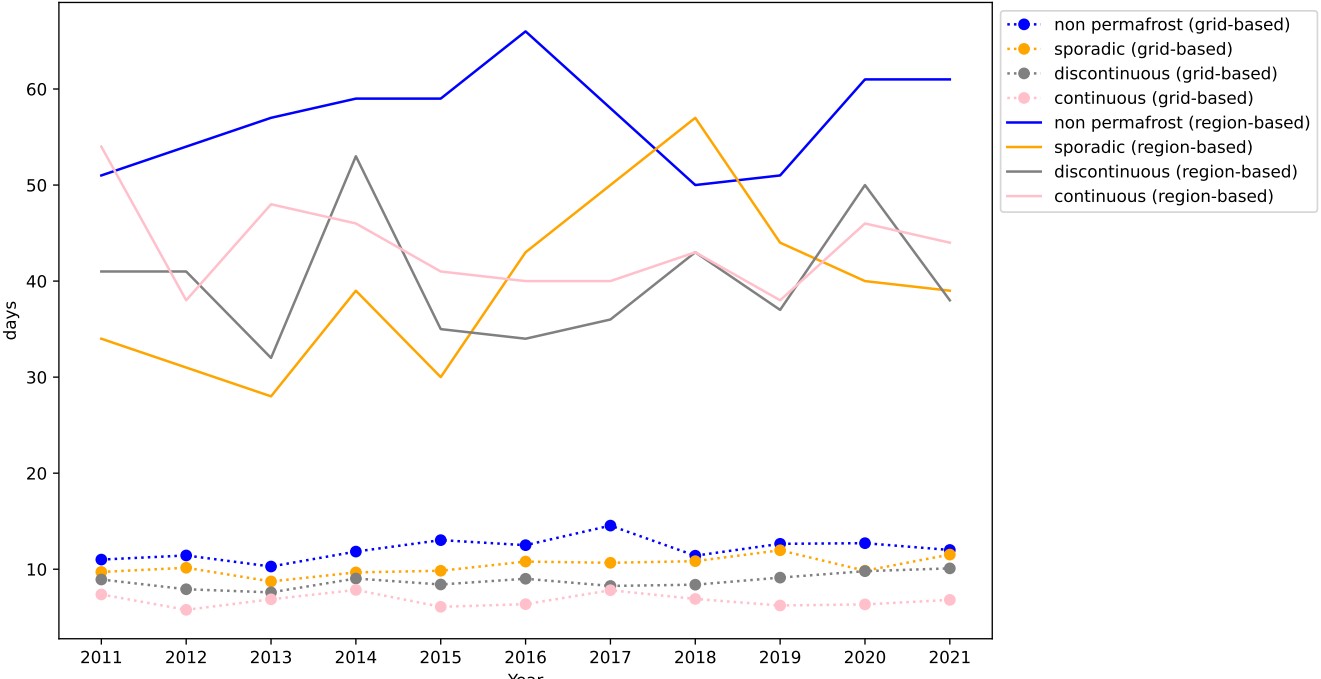

**Figure 2.** The length of the melting season defined with the two methods in the northern high latitude permafrost zones. The dotted lines represent the grid-based mean length of the melting season. The solid lines represent the region-based melting season. The size of grid-cells in the north-south direction was taken into account when defining the melting season with both methods.

With both methods, the season typically started and ended the earliest in the southernmost regions and the latest in the northernmost regions, with the season starting later in some of the southernmost grid cells (mostly mountainous regions). In the southernmost regions, the season started earliest in January, but the average start of the season in the non-permafrost zone was in April. In the continuous permafrost zone, the season started earliest in February but the average start of the melting season was in May. In the non-permafrost zone, the season ended earliest in January and on average in April. In the continuous permafrost zone, the season ended earliest in March and on average in June. With the region-based approach in the non-permafrost region, the melting season started in the middle of March and ended in the beginning of May. In the sporadic region, the melting season started in the middle of April and ended in the middle of May. In the discontinuous permafrost region, the melting season started in the second half of April and ended at the end of May. In the continuous permafrost region, the melting season started in the beginning of May and ended in the middle of June. In the Eurasian continent, the melting season started earlier in the west and later in the east. The regions with the most permafrost are located in the eastern part of Eurasia which means that the regions with less permafrost started to melt first. In the American continent, an east–west thawing gradient is not apparent. However, the melting season started and ended the latest where the most permafrost is located.





**Figure 3.** Grid-based (figures a,b and c) and region-based (figures d,e and f) average length, start day, and end day of the melting season in the northern high latitudes, averaged over 2011–2021. Notice the different color-range in the melting season figures on the left side. In the region-based figures, only the illustrated length, start and end days are ticked on the color-bar. The color range in the start and end day figures are the same in the grid- and region-based figures.

### 3.1.2 Melting season in the Hudson Bay lowlands and the Western Siberian lowlands

The melting season was additionally studied in the Hudson Bay lowlands (land area in 50–60° N latitudes and 75–96° W longitudes) and the Western Siberian lowlands (land area in 52–74° N latitudes and 60–94.5° E longitudes). Hudson Bay





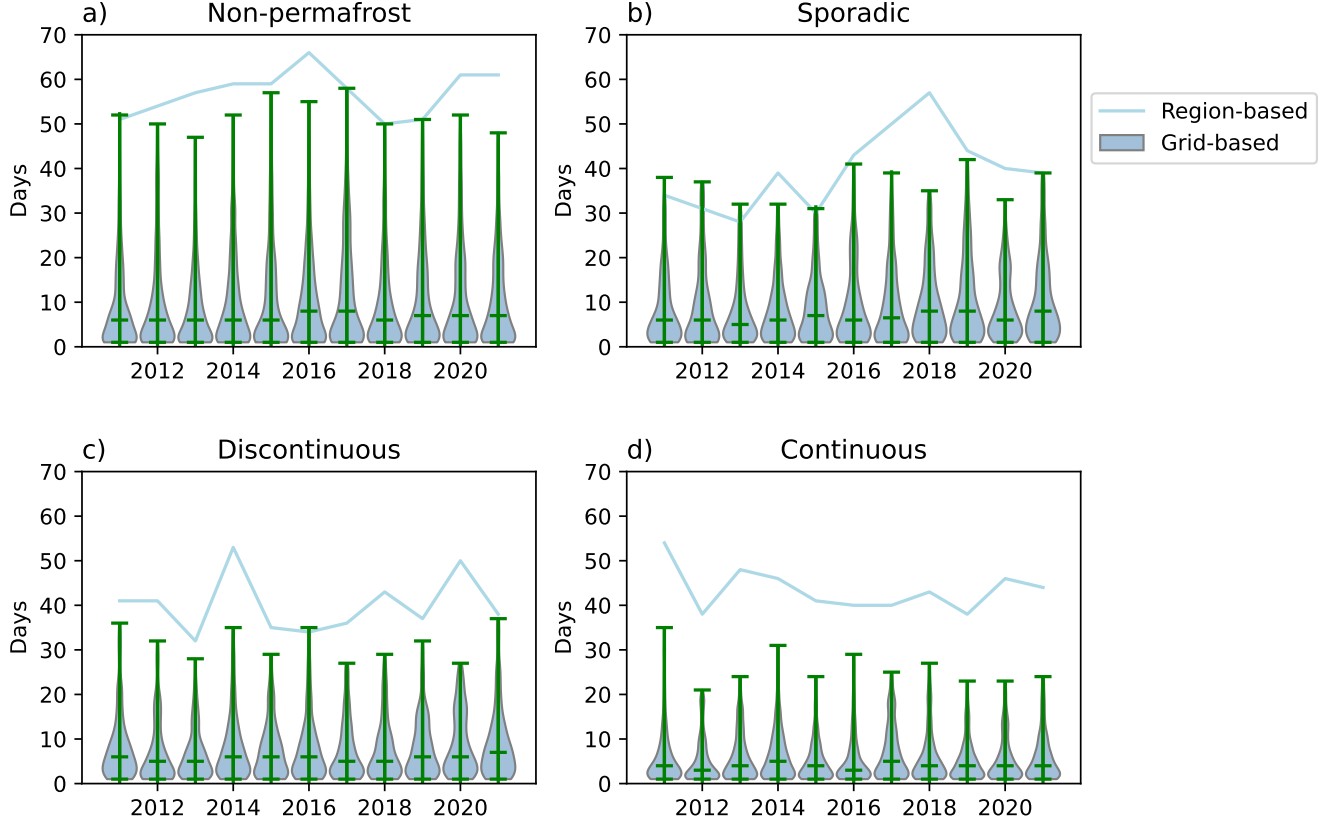

**Figure 4.** The length of the melting season defined with the two methods in the northern high latitude zones: a) non-permafrost, b) sporadic, c) discontinuous and d) continuous permafrost. The green violin plots represent the grid-based length distribution in the four zones. The medians as well as the min and 97,5 percentile max values for each year are shown as well with the green lines. Grid cells where the length of the melting season was longer than the 97,5 percentile have been masked out. The blue lines represent the region-based length of the season.

lowlands and Western Siberian lowlands are some of the largest methane emitting wetlands in the northern high latitudes. They consist of the four permafrost zones, except the Hudson Bay lowlands where there is no continuous permafrost. On average, the

grid-based melting season was slightly shorter in the Western Siberian lowlands (∼9 days) than in the Hudson Bay lowlands (∼10 days). The season started and ended a few days earlier on average in the Western Siberian lowlands than in the Hudson Bay lowlands. The region-based melting season length was not defined separately for these wetland regions, but the original region-based lengths in the four permafrost zones were used.

      Overall, in the Hudson Bay lowlands, there was a negative correlation between the mean temperature and mean length of

the melting season ($p < 0.05$ in all zones) with the grid-based approach. In the Western Siberian lowlands, there was also a negative correlation in all of the permafrost zones but it was statistically significant only in the non-permafrost and sporadic





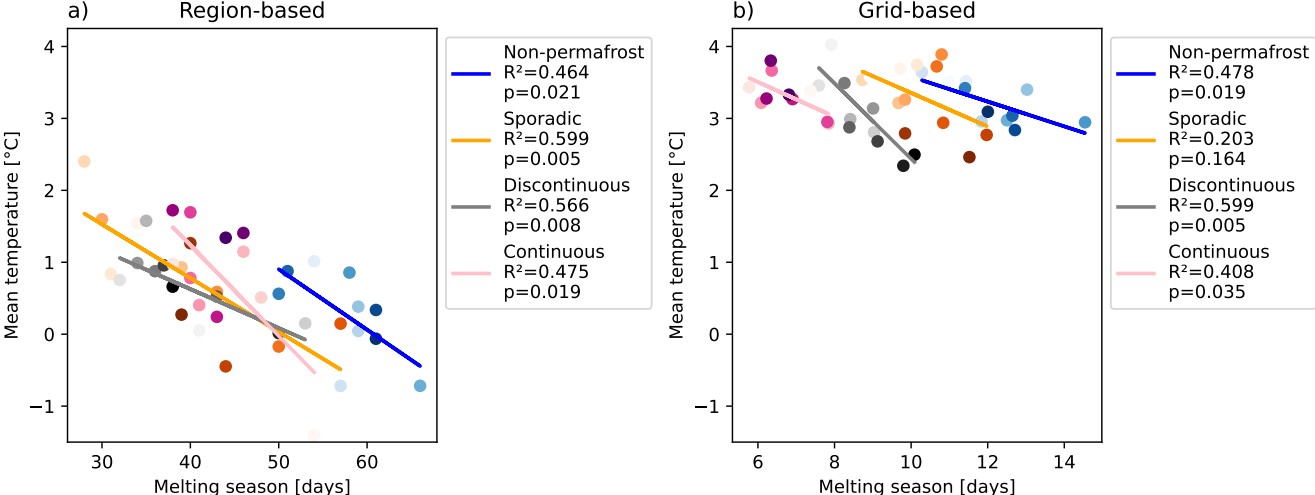

**Figure 5.** Figure a depicts the region-based relationship between the length of the melting season and the mean temperature of the melting season. Figure b depicts the grid-based relationship between the mean length and the mean temperature of the melting season. The scatter plot color gradient represents the different years with 2011 being the lightest color and 2021 being the darkest. $R^2$ and p in the legends are the coefficient of determination and p-values of the slopes from linear regression fit, indicating statistical significance of the coefficient of determination.

zones ($p < 0.05$). This means that the season was shorter when the mean temperature was higher at least with the grid-based approach.

With the region-based approach, there was not as clear negative correlation between the length of the melting season and
the mean temperature of the melting season especially in the Western Siberian lowlands ($p < 0.05$ only in the sporadic zone). In the Hudson Bay lowlands, there was a statistically significant negative correlation in the sporadic and discontinuous zones ($p < 0.05$). If the melting season was defined with the region-based approach separately in the Hudson Bay and the Western Siberian lowlands areas, then the correlations might have been more significant.

### 3.2 Melting season methane emissions

The average annual region-based melting season emissions in the whole northern high latitude region in our study were $1.83 \pm 0.27$ Tg (Table A3), where $\pm$ is the maximum difference from the mean here and hereafter, and grid-based annual melting season emissions were $0.45 \pm 0.10$ Tg (Table A4). The year-to-year variation was modest, within 15% of the average emissions with the region-based approach and 23 % of the average emissions with the grid-based approach. The grid-based emissions were much smaller than the region-based emissions in all the zones due to shorter melting seasons on average (see
Section 3.1). The melting season methane emissions were the largest in the southernmost zone, non-permafrost, which was also the largest zone, and smallest in the discontinuous permafrost zone (Fig. 6).





To study which variables were linked to melting season methane emissions, the relationship between the length of the melting season and the methane emissions was studied (Fig. 8). With both methods, the correlation between the length of the melting season and the total melting season emissions was positive in all the permafrost zones, meaning that the melting season had higher emissions when the season was longer. The positive correlation can be seen most clearly in the non-permafrost zone. The correlations were statistically significant ($p < 0.01$) in all the zones with both methods.

Additionally, the emission rate, as the units of µg(CH$_4$) m$^{-2}$ s$^{-1}$, was calculated for the different zones. The region-based mean emission rate was defined by dividing the total emissions with the area of the zone and the length of the melting season. The grid-based mean emission rate was defined as the average emission rate between the different grid cells. The correlation between the rate of emissions and the length of the melting season was not as clear as between the total melting season emissions and length, especially with the grid-based approach ($p > 0.05$ in all zones). There was a small negative correlation in the sporadic and discontinuous zones and positive ones in the other two zones. With the region-based approach (Fig. 8), there was a stronger negative correlation in the sporadic zone between emission rate and length of the season ($p < 0.01$). In other zones, the correlation between emission rate and length was insignificant ($p > 0.05$). The negative correlation in the sporadic zone could exist because of the negative correlation between the length and temperature of the melting season, and therefore the temperature would correlate positively with the emission rate. This relationship was studied in the grid-based Fig. A5, where grid cells with a negative emission rate were masked out, but no correlation was found in any of the zones. The region-based relationship between the variables was very similar. This indicates that the total emissions grew when the season was longer but there was no clear indication that the emission rate would be stronger or weaker when the season is longer.

Average grid-based emissions in the Hudson Bay lowlands were $0.03 \pm 0.01$ Tg and in the Western Siberian lowlands they were $0.11 \pm 0.04$ Tg (Table A5). The region-based emissions in the Hudson Bay lowlands were $0.10 \pm 0.03$ Tg and $0.47 \pm 0.14$ Tg in the Western Siberian lowlands. When the contribution of different permafrost zones in the Hudson Bay lowlands and the Western Siberian lowlands were studied, most of the grid-based emissions in these areas came from the non-permafrost zone during most of the years. In 2012 and 2016 in the Hudson Bay lowlands, most emissions came from the sporadic zone, instead. With the region-based approach, most emissions came from the non-permafrost zone every year in both Hudson Bay lowlands and Western Siberian lowlands. The non-permafrost zone was the largest permafrost zone in both regions.

The years with the highest emissions in the entire northern latitude regions were 2011, 2014–2017, and 2019 (Tables A3, A4). With the region-based approach, the year with the largest emissions was 2014 and with the grid-based approach, it was 2017. Maps of some of the highest emitting melting seasons in the Hudson Bay lowlands and Western Siberian lowlands are shown in Figures 7 and A6. We can see that some of the grid-cells with the highest emissions were located inside these two lowlands. However, the years with highest emissions in the Hudson Bay lowlands and Western Siberian lowlands differed from the years with highest emissions of the whole northern latitude regions. On the other hand, the years with the highest emissions in the Hudson Bay lowlands and Western Siberian lowlands matched between the two methods. The highest grid-based emissions in the Hudson Bay lowlands were found in 2012 and 2020, and region-based in 2012 and 2014. In the Western Siberian lowlands, the highest emissions were in 2014 and 2015 with both approaches. The highest emissions of the Western





Siberian lowlands align better with the overall highest emitting years, likely due to the larger area of the Western Siberian lowlands to the Hudson Bay lowlands. However, there were not major differences in the emissions between the years in both regions. The average grid-based melting season length varied between the permafrost zones during the years with the most emissions in the Hudson Bay lowlands and the Western Siberian lowlands. Usually the season was longer than average at least in one of the four regions in both lowlands during those years. In the Hudson Bay lowlands during 2012 and 2020, the melting season was usually longer in a region where the mean temperature of the melting season was also colder than average. Same was seen in the Western Siberian lowlands in 2014.

The region-based melting season was not longer than the average in any of the zones in 2012, but in 2014, it was longer than the average in all the zones, except in the sporadic zone. In 2015, it was only longer than average in the non-permafrost zone. However, the region-based melting season was defined from the whole permafrost zone instead of only the area within Hudson Bay lowlands and Western Siberian lowlands, which might explain why the melting seasons were not exceptionally long during the years with highest emissions in these two regions. Both regions were also mostly covered by the non-permafrost zone, which explains why the emissions were generally larger when the melting season was longer in the non-permafrost zone.





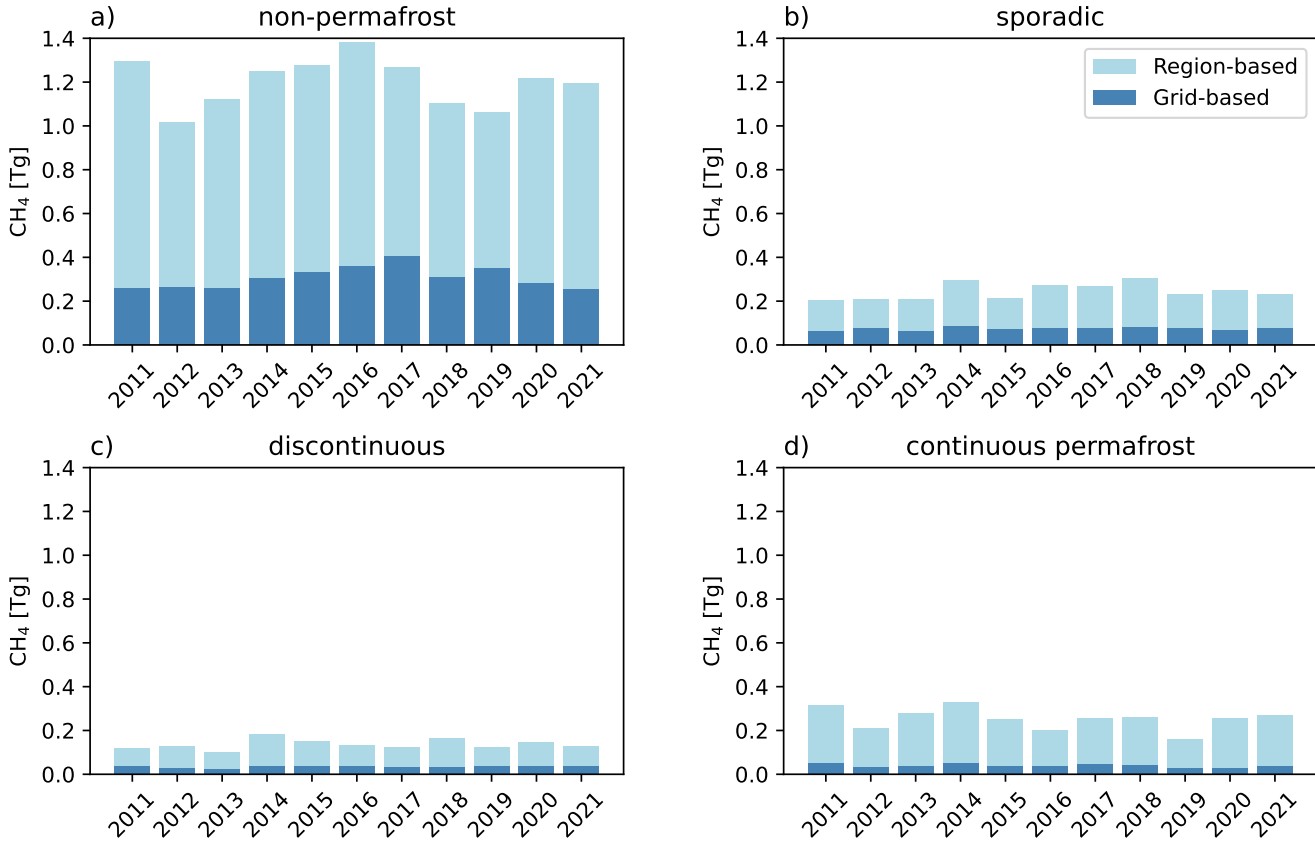

**Figure 6.** Total emissions in the four permafrost zones: a) non-permafrost, b) sporadic, c) discontinuous and d) continuous permafrost. The region-based emissions are coloured in blue and the grid-based emissions in green.





**Figure 7.** Grid-based emission maps of the melting season in the years that had some of the highest emissions in the Hudson Bay lowlands and Western Siberian lowlands. Hudson Bay lowlands have been outlined with red borders, and Western Siberian lowlands with blue borders.



**Figure 8.** Figures a and b depict the region-based relationship between the total emissions (a) or the emission rate (b) and the length of the season. Figures c and d depict the grid-based relationship between the total emissions (c) or the mean emission rate (d) and the mean length of the melting season. Notice the different value range on the y-axis in a and c. The scatter plot color gradient represents the different years with 2011 being the lightest color and 2021 being the darkest. $R^2$ and p in the legends are the coefficient of determination and p-values of the slopes from linear regression fit, indicating statistical significance of the coefficient of determination.



## 4 Discussion

### 4.1 The melting season

#### 4.1.1 Definition of the melting season and the SMOS F/T algorithm

The spring melting season was defined for the northern high latitude wetlands for the first time based on the SMOS F/T data. Because the SMOS F/T algorithm cannot distinguish between thawed soil and wet snow, it likely first detects the melting of snow instead of thawing of the soil in springtime. Thus, the defined melting seasons probably already started when the snow began to melt and before the soil thawing and made the season longer than the actual soil thawing lasted. However, methane emissions can start already during snow melting and cracking of the soil, when liquid water can seep to the soil (Rinne et al., 2007; Raz-Yaseef et al., 2017) and wintertime methane reserves are released. Hence, it is justified to define the melting season to start from the melting of the snow. In addition, including either snow depth data from in-situ measurements or fractional snow cover data from satellites could have helped our algorithm to detect the starting of the soil thawing.

The amount of SMOS F/T pixels in a $1° \times 1°$ grid cell made the definition of melting season spatially inconsistent. The number of pixels decreased towards the north, requiring a higher fraction of thawed soil in northern grid cells before the melting season began. However, with our definition, the absolute area, which had to be thawed before the season could start, was similar across the study area, though it also meant that the estimated thawing state of the soil was more uncertain if there were missing SMOS F/T datapoints. The difference in the amount of pixels in a grid cell might have introduced systematic biases, causing longer melting seasons in south and shorter in north.

With both methods, the melting season generally started the earliest in the southern regions. However, there were some mountainous regions in the south, for example, the mountains in Mongolia, and the Rocky Mountains in the western coast of North America (Fig. 3), where the season started and ended later, causing long melting seasons. This is likely due to higher altitude and lower temperatures. The SMOS F/T data is not as reliable in the mountainous regions as it is elsewhere, mainly because there is less soil substance, especially at higher elevations. Since the SMOS satellite measures the soil freezing through the permittivity difference between ice and water, it does not measure the soil freezing correctly in areas with less absorbent soil. Thus, the melting of snow or ice on solid rock does not significantly affect what is measured. A sufficient difference between thawed and frozen states requires soil that has absorbed water. Additionally, the topography affects the remote sensing measurements: the measurement angle varies a lot in the mountains, which means that with different measurement angles, the measured thawing state of the soil also changes.

#### 4.1.2 Region-based and grid-based melting season

The region-based melting season was on average about 4 times as long as the grid-based melting season. The region-based approach might not represent the local melting season accurately because of the substantial area of the permafrost zones and the difference between the start and end days inside one zone was large. In other words, the melting season in the southern grid cells might have already ended, while soil was still frozen in northern parts of the zone. However, with this method, the





thawing fraction varied less during the melting season, and the beginning and end of the melting season were more consistent, making it easier to compare the different permafrost zones' melting seasons and methane emissions with each other.

Each of the four permafrost zones consisted of areas in the North American and the Eurasian continents. Especially the region-based melting season could have been more precise if it was defined separately for the two continents due to their different year-to-year variability of climates.

The grid-based melting season was on average almost twice as long in the southernmost zone: non-permafrost (12 days) compared to the northernmost permafrost zone: continuous permafrost (7 days). This suggests that the season was shorter when there was more permafrost and the further north the region was, except for the mountainous permafrost regions in the south (Fig. 3). In a small area, the soil probably melts simultaneously leading to a shorter melting season. Additionally, the season could not start before the minimum thawing value was surpassed for the last time in the spring. This meant that for some grid cells, there might have been variation in the thawing value before this date, and this variation could have been included in the melting season. However, with this method the spring melting season was separated from the autumn freezing season if it continued past the turn of the year.

### 4.1.3 Timing and length of the melting season

Different methods used to study the timing of snowmelt and spring thawing season have produced varying results. According to the definitions of the Finnish Meteorological Institute (FMI) based on temperature data, spring starts approximately in the middle of March in southern Finland and in April in northern Finland (Ilmatieteen laitos, 2025) and lasts for two to four weeks during which snow melts and the growing season starts. Our estimated melting seasons in Finland, which is located mostly in the non-permafrost area, coincide with this: on average, the melting season started in March–April using both the grid-based and the region-based methods.

According to regional studies in northern Alaska, underlain by continuous permafrost, the spring thawing season in the tundra region was approximately 20 days long (Zona et al., 2016; Bao et al., 2020), which is shorter than the region-based melting season defined here (45 days on average, Table A2) and longer than the average grid-based melting season (10 days). With our grid-based method, the average length of the melting season during our study period in the grid-cells closest to the coordinates of the measurement stations used by Zona et al. (2016) and Bao et al. (2020) varied between 4-19 days. Overall, in many grid-cells in the whole northern high latitude domain, the length of the melting season was ~20 days. Both Zona et al. (2016) and Bao et al. (2020) used soil temperature data to define the thawing of the soil. We did not use in-situ soil temperature data in this study as in-situ measurements of frozen soil are rare, and thus not suitable for our larger region study. Other option would have been to use reanalysis soil temperature data such as ERA5, which has shown higher skills than other products and a significant improvement over its predecessor (Li et al., 2020). However, it is not well-suited for permafrost research (Cao et al., 2020). Thus, SMOS F/T data is a good proxy for the soil thawing.

In the study by Tenkanen et al. (2021), the SMOS F/T data showed later soil freezing but earlier soil thawing than a process-based ecosystem model, resulting in a shorter winter season. The region-based season instead was longer. This might be due to the SMOS F/T data starting the melting season already from the melting of the snow instead of the soil. According to our prior





LPX–Bern DYPTOP v1.4 (Fig. A7), the soil temperature rose to 0 °C during the region-based melting season on all permafrost zones. This means that the SMOS F/T melting season started earlier than the process-based ecosystem model melting season. However, the prior had a monthly temporal resolution compared to the daily resolution of the SMOS F/T data, and so the soil temperature rising above 0 °C in the middle of the melting season is a good indicator for the correct timing of our melting season.


According to the prior emissions used in this study (LPX–Bern DYPTOP v1.4), peat emissions were relatively high during the spring melting season than emissions from inundation for all other permafrost zones but continuous permafrost (Fig. A7). However, even in the continuous permafrost zone, the peat emissions are higher than inundation in April, and are rising during the melting season. At the beginning of spring, extensive snow melt inundation causes a large part of methane emissions, and

after the end of the melting season, the methane emissions are dominated by peat emissions. Even though, methane emissions from peat dominated over emissions caused by inundation for a large part of the year, there were evident peaks in inundation emissions in every permafrost zone during and/or right after the melting season. This indicates that the melting season timing we defined is reasonable.

### 4.2 Methane emissions during the melting season

The way we defined the melting season also affected the estimated melting season methane emissions. The emissions were optimized weekly, which were interpolated linearly to daily values from one optimized weekly value to the next weekly value before the calculation of the melting season emissions. This introduced a new source of uncertainty and affected the estimated melting season methane emissions, especially when using the grid-based approach, as the length of the grid-based season was much shorter and often less than a week (between 51% in the non-permafrost zone and 70% in the continuous permafrost

zone).

In the Hudson Bay lowlands and Western Siberian lowlands, as wells as in the four permafrost zones, the methane emissions were distributed unevenly (Fig. 7). This might be explained by the distribution of wetlands in the lowlands, as wetlands typically have higher methane emissions than, e.g., forests, which might even be a sink of methane. The years with some of the highest emissions were distributed evenly across the study period (Fig. 7), which means that there is no direct indication of increasing

melting season emissions.

### 4.2.1 Methane bursts during the spring melting season

Multiple previous studies have shown higher emission rates to our results with many of them showing large bursts of $CH_4$ from wetlands during the spring thawing season (Jin et al., 1999; Tokida et al., 2007; Song et al., 2012; Tagesson et al., 2012; Mastepanov et al., 2013; Zona et al., 2016; Raz-Yaseef et al., 2017; Bao et al., 2020). These events typically last for a short

time, from few hours to a few days (Song et al., 2012). Such emission bursts could have caused large emissions even during a shorter melting season, causing the relationship between the length and the emissions of the melting season to be non-linear. However, we found a positive correlation between the total melting season methane emissions and length, which was observed with both the grid-based and the region-based methods in permafrost zones and Western Siberian and Hudson Bay lowlands.





This indicates that the bursts were not large enough to be detected with our model resolution ($1° \times 1°$) and weekly temporal

optimization of the fluxes. The emission rates were also estimated from the whole area of the $1° \times 1°$ grid-cell instead of the wetland area, which made the emission rates smaller compared to the local studies. Another reason for the positive correlation between the values could be that in our study, we focused to study emissions within larger permafrost zones. In individual grid-cells, some emission peaks could have been detected but that was not explicitly studied here. Above mentioned studies reporting large $CH_4$ bursts during the spring thaw used field measurements to define the emission rate which have a sensitivity

for a smaller areas compared to our inverse model. Emission rates defined from the eddy covariance measurement measure the local methane bursts rather than the average rates from a larger permafrost zone.

Using a process-based model Castro-Morales et al. (2018) showed an emission rate closer to what we found in this study. The atmospheric inverse model used here used atmospheric $CH_4$ measurements to inform and "re-evaluate" the process-based estimates. In most years, our inversion showed an increase from the prior emission estimates which were based on the process

model LPX-Bern DYPTOP. This means that there might have been short-lived emission bursts during the spring time, which the process model was not able to produce due to the poor spatial and temporal resolution or missing processes, or that the process model emission estimates were overall too low during the spring seasons.

With our inverse model, the grid-based emission rates were higher in some grid cells (Fig. A5), and on average, the grid-based emission rates were higher than the region-based (Fig. 8). The grid-based melting season likely represent local emission

bursts better than the region-based approach. Inside one permafrost region, the fluxes vary a lot, as the whole area is not permafrost or wetlands. This lowers the region-based emission rate. The local measurement studies might be more comparable to the spring melting season emissions in the Hudson Bay lowlands and Western Siberian lowlands in grid cells, where the total area of the cell is mostly wetland. However, the focus of this study was to estimate the total melting season emissions during the springtime rather than occasional hotspots.

### 4.2.2 Magnitude of the spring melting season methane emissions

According to Saunois et al. (2025), the annual global methane emissions estimated from atmospheric inversions were 575 [553–586] Tg $CH_4$ yr$^{-1}$, in 2010–2019. Of these emissions, 165 [145–214] Tg $CH_4$ yr$^{-1}$ were from wetlands. The mean global biospheric emissions using the CTE-$CH_4$ data from this study for the years 2011–2021 were 124 Tg $CH_4$ yr$^{-1}$, including also soil sink (LPX-Bern DYPTOP prior soil sink 33 Tg $CH_4$ yr$^{-1}$). In the northern high latitudes (defined in the chapter 2.3),

the average annual methane emission were 23 Tg $CH_4$ yr$^{-1}$. The region-based spring melting season emissions were 1.83 Tg (8.1% of the annual northern high latitude emissions) and grid-based were 0.45 Tg (2% of the annual northern high latitude emissions), only a small portion of the annual total global emissions. The length of the melting season with the grid-based method was 3%, and with the region-based method 12%, of the total length of a year.

Methane emissions during other seasons and the high northern latitudes have been studied previously. According to Tenkanen

et al. (2021), the winter cold season (November to April) emissions were 3.3 Tg in the northern high latitudes (above 50° N), which included partly emissions from both the autumn freezing and spring thaw seasons. The autumn freezing season methane emissions from the same model with a different setup were 1 Tg (Tenkanen, 2019), which means that the region-based spring




melting season emissions were a bit larger than the autumn emissions, and grid-based were smaller. Erkkilä et al. (2023) estimated the autumn freezing, winter cold season and summer thaw season emissions in the northern high latitudes with the

same inverse model but a different setup to ours. They found emissions to be 1.2 Tg in winter, 0.73 Tg in the freezing period, and 16.2 Tg in the thaw (summer) period. Both winter and freezing season emissions were smaller than our region-based spring emissions but larger than the grid-based emissions.

According to a study which used upscaled flux measurements, the spring methane emissions during the melting of the soil in wetlands in a northern high latitude permafrost region were 0.5–0.97 Tg in 2011 (Song et al., 2012). In a study by Ito et al.

(2023), the spring season (March–May) methane emissions were calculated with multiple process-based ecosystem model models for the northern wetlands (>45°), and their mean value was $0.80 \pm 1.89$ Tg $CH_4$ $yr^{-1}$ ($3.07 \pm 9.61$ % of the annual emissions), where the error is the variation between the maximum and minimum model result. These values are closer to the grid-based emissions estimated in this study compared to the region-based emissions.

Defined from the inverse model used in this study, the mean annual emissions in Hudson Bay lowlands and Western Siberian

lowlands were the size of 2.9 Tg and 5.0 Tg, respectively. This is close to the values defined in other studies (Thompson et al., 2017; Peltola et al., 2019; Tenkanen et al., 2021). The average region-based melting season emission in the Hudson Bay lowlands and the Western Siberian lowlands were the size of 0.1 Tg and 0.47 Tg, respectively. The grid-based emission were smaller with the magnitude of 0.03 Tg and 0.11 Tg, respectively. This means that together they produced approximately 31% of the total melting season emissions from the northern high latitude wetlands with both methods. This is quite a significant part

of the total melting season emissions when their areas only represent a portion of the total study region (Hudson Bay lowlands is 2.5% and Western Siberian lowlands is 9.6% of the total study region). The region-based emissions were 7% and grid-based emissions were 1.7% of the annual emissions from the Hudson Bay lowlands and the Western Siberian lowlands. However, the emissions in the Western Siberian lowlands could have been overestimated, because the NIES observation sites used in the inversion had a different calibration scale (3.0 to 5.5 ppb higher) than WMO CH4 X2004A scale as in the ObsPack (Sasakawa

et al., 2025). The NIES observations are now included in the newer version of ObsPack, with scale corrected. In conclusion, the Western Siberian lowlands and Hudson Bay lowlands regions produced a large portion of the total melting season emissions in the northern high latitudes but these emissions represented only a small portion of the total annual emissions from these regions.

## 5    Summary and conclusion

The spring melting season was defined for the northern high latitude wetlands with the SMOS F/T soil thawing data separately for four permafrost zones: seasonally frozen non-permafrost, sporadic, discontinuous and continuous permafrost. The melting season was defined separately with a region-based and a grid-based approach. The region-based method is comparable to studies with monthly emission estimates while the grid-based method is more comparable to local studies. The region-based season started and ended the earliest in the most southernmost permafrost zone and latest in the northernmost zone. With

the grid-based approach, there was more variation in the start and end days of the season in the north-south direction mostly



because of the mountainous regions. The region-based melting season lengths varied more from year-to-year than the grid-based seasons. With both methods, the non-permafrost zone had the longest season on average. The average grid-based melting season was shortest in the continuous permafrost zone and the region-based was shortest in the sporadic and discontinuous zones. We found that the length of the melting season was dependent on the average 2 m temperature of the melting season, with a longer season having a lower mean temperature. This could have been caused by the early onset of melting, followed by variation between temperatures, making the mean temperature of longer seasons lower. We found the SMOS F/T data to be useful in estimating spring melting season, and for example, using it to inform process-based models to account for soil freeze/thaw state could lead to better constrained methane emission estimates.

In the four permafrost zones, the melting season methane emissions were the largest in the non-permafrost zone and the smallest in the discontinuous zone with both methods. The total region-based melting season emissions for the four zones were $1.83 \pm 0.27$ Tg $CH_4$ per season and grid-based emissions were $0.45 \pm 0.10$ Tg $CH_4$ per season. These emissions represented 8.1 % and 2 % of the total annual northern high latitude emissions according to our inverse model results (23 Tg). We found that the total melting season emissions depended on the length of the melting season, with more methane emitted during longer seasons.

We additionally studied the emissions in the Hudson Bay lowlands and Western Siberian lowlands, as they are the largest wetlands in the northern high latitudes. Their melting season methane emissions were a significant portion of the total melting season emissions (31% with both methods and both lowlands combined) but only a small part of the total annual northern high latitude emissions (2.5% with the region-based approach and 0.6% with the grid-based approach). The Western Siberian lowlands had higher emissions than the Hudson Bay lowlands, even though it had a slightly shorter grid-based melting season on average than the Hudson Bay lowlands but with a lower average mean temperature with the grid-based approach.

The total melting season emissions in the northern high latitudes were only a small portion of the total annual emissions. However, the melting season emissions could grow in the future due to climate change and the thawing of permafrost. On the other hand, our results showed that a shorter melting season had smaller total emissions and that a shorter season had a higher mean temperature on average. Increasing temperatures could lead to shorter melting seasons and lower melting season methane emissions but also a longer thaw season. Additionally, it is still unclear how feedback loops will affect emissions and the thawing of the permafrost. To get a better look at the spring melting season emissions, the results from several inversions could be compared to each other. In addition, going further to the level of individual wetlands and comparing their flux measurements in the springtime could bring more clarity to true wetland emissions.

*Data availability.* The data processed for this study will be available on the FMI Research Data Repository METIS (https://fmi.b2share.csc.fi/, DOI:doi will be added here)

**Appendix A**



**Table A1.** List of surface observation sites used in inversions. Observation Uncertainty (Obs. Unc.) is used to define diagonal values in the observation covariance matrix. The data type is categorized into two measurements (discrete (D) and continuous (C)).

| site code | Site Name | Site Country | Contributor | Site Longitude [°E] | Site Latitude [°N] | Altitude* [m a.s.l.] | Obs. Unc. [ppb] | Data Type D/C | Dates min. [year/month] | Dates max. [year/month] |
|---|---|---|---|---|---|---|---|---|---|---|
| ABP | Arembepe, Bahia | Brazil | NOAA | -38.17 | -12.76 | 6.00 | 4.5 | D | 2006/10 | 2010/01 |
| ABT | Abbotsford, British Columbia | Canada | EC | -122.34 | 49.01 | 93.00 | 30.0 | C | 2014/03 | 2020/12 |
| ALT | Alert, Nunavut | Canada | NOAA | -62.51 | 82.45 | 190.00 | 15.0 | D | 2000/01 | 2021/10 |
| ALT | Alert, Nunavut | Canada | EC | -62.51 | 82.45 | 195.00 | 15.0 | C | 2000/01 | 2020/12 |
| AMY | Anmyeon-do | Republic of Korea | NOAA | 126.33 | 36.54 | 87.00 | 30.0 | D | 2013/12 | 2021/11 |
| ARA | Arcturus | Australia | CSIRO | 148.47 | -23.86 | 185.00 | 15.0 | D | 2010/05 | 2013/10 |
| ASC | Ascension Island | United Kingdom | NOAA | -14.40 | -7.97 | 90.00 | 15.0 | D | 2000/01 | 2021/11 |
| ASK | Assekrem | Algeria | NOAA | 5.63 | 23.26 | 2715.00 | 25.0 | D | 2000/01 | 2019/11 |
| AZR | Terceira Island, Azores | Portugal | NOAA | -27.36 | 38.76 | 24.00 | 15.0 | D | 2000/01 | 2021/09 |
| AZV | Azovo | Russian Federation | NIES | 73.03 | 54.71 | 190.00 | 30.0 | C | 2009/10 | 2018/12 |
| BAR | Baranova | Russian Federation | FMI | 101.62 | 79.28 | 30.00 | 4.5 | C | 2015/11 | 2021/02 |
| BCK | Behchoko, Northwest Territories | Canada | EC | -115.92 | 62.80 | 220.00 | 15.0 | C | 2010/10 | 2020/12 |
| BHD | Baring Head Station | New Zealand | NOAA | 174.87 | -41.41 | 90.00 | 4.5 | D | 2002/03 | 2021/10 |
| BIK | Bialystok | Poland | MPI-BGC | 23.01 | 53.23 | 483.00 | 25.0 | C | 2005/07 | 2014/06 |
| BIR | Birkenes | Norway | NILU | 8.25 | 58.39 | 225.00 | 25.0 | C | 2014/01 | 2014/01 |
| BKT | Bukit Kototabang | Indonesia | NOAA | 100.31 | -0.20 | 875.00 | 75.0 | D | 2004/01 | 2021/11 |
| BLK | Baker Lake, Nunavut | Canada | EC | -96.01 | 64.33 | 61.00 | 15.0 | C | 2017/07 | 2019/11 |
| BME | St. Davids Head, Bermuda | United Kingdom | NOAA | -64.65 | 32.37 | 17.00 | 15.0 | D | 2000/01 | 2010/01 |
| BMW | Tudor Hill, Bermuda | United Kingdom | NOAA | -64.88 | 32.26 | 33.00 | 15.0 | D | 2000/01 | 2021/12 |
| BRA | Bratt's Lake Saskatchewan | Canada | EC | -104.71 | 50.20 | 630.00 | 75.0 | C | 2009/10 | 2020/12 |
| BRW | Barrow Atmospheric Baseline Observatory | United States | NOAA | -156.61 | 71.32 | 27.46 | 15.0 | C | 2000/01 | 2021/12 |
| BRW | Barrow Atmospheric Baseline Observatory | United States | NOAA | -156.58 | 71.32 | 16.00 | 15.0 | D | 2000/01 | 2021/11 |
| BRZ | Berezorechka | Russian Federation | NIES | 84.33 | 56.15 | 248.00 | 75.0 | C | 2008/05 | 2018/12 |
| BSD | Bilsdale | United Kingdom | UNIVBRIS | -1.15 | 54.36 | 628.00 | 30.0 | C | 2014/01 | 2019/12 |
| CBA | Cold Bay, Alaska | United States | NOAA | -162.71 | 55.21 | 25.00 | 15.0 | D | 2000/01 | 2021/11 |
| CBY | Cambridge Bay, Nunavut Territory | Canada | EC | -105.06 | 69.13 | 47.00 | 15.0 | C | 2012/12 | 2020/12 |
| CFA | Cape Ferguson | Australia | CSIRO | 147.06 | -19.28 | 5.00 | 25.0 | D | 2000/01 | 2021/04 |
| CGO | Cape Grim, Tasmania | Australia | NOAA | 144.68 | -40.68 | 164.00 | 4.5 | D | 2000/01 | 2021/10 |
| CGO | Cape Grim | Australia | CSIRO | 144.68 | -40.68 | 94.00 | 15.0 | C | 2012/07 | 2021/07 |
| CGR | Charles Point, Darwin | Australia | CSIRO | 12.65 | 37.67 | 9.00 | 25.0 | C | 2015/04 | 2018/12 |
| CHL | Churchill, Manitoba | Canada | EC | -93.82 | 58.74 | 89.00 | 15.0 | C | 2011/10 | 2020/12 |
| CHM | Chibougamau, Quebec | Canada | EC | -74.34 | 49.69 | 423.00 | 30.0 | C | 2007/08 | 2011/04 |
| CHR | Christmas Island | Republic of Kiribati | NOAA | -157.15 | 1.70 | 5.00 | 15.0 | D | 2000/01 | 2020/01 |
| CPS | Chapais,Quebec | Canada | EC | -74.98 | 49.82 | 431.00 | 15.0 | C | 2011/12 | 2020/12 |
| CPT | Cape Point | South Africa | NOAA | 18.49 | -34.35 | 260.00 | 25.0 | D | 2010/02 | 2021/10 |
| CRI | Cape Rama | India | CSIRO | 73.83 | 15.08 | 66.00 | 75.0 | D | 2000/01 | 2013/01 |
| CRV | Carbon in Arctic Reservoirs Vulnerability Expe... | United States | NOAA | -147.60 | 64.99 | 643.13 | 15.0 | C | 2011/10 | 2021/12 |
| CRZ | Crozet Island | France | NOAA | 51.85 | -46.43 | 202.00 | 4.5 | D | 2000/01 | 2021/01 |
| CUR | Monte Curcio | Italy | IIA | 16.42 | 39.32 | 1801.00 | 15.0 | C | 2014/12 | 2017/12 |
| CYA | Casey Station, Antarctica | Australia | CSIRO | 110.52 | -66.28 | 55.00 | 4.5 | D | 2000/01 | 2021/01 |
| DEM | Demyanskoe | Russian Federation | NIES | 70.87 | 59.79 | 155.00 | 30.0 | C | 2005/09 | 2018/12 |
| DRP | Drake Passage | Drake Passage | NOAA | -61.68 | -59.07 | 10.00 | 4.5 | D | 2006/03 | 2021/05 |
| DSI | Dongsha Island | Taiwan | NOAA | 116.73 | 20.70 | 8.00 | 15.0 | D | 2010/03 | 2021/10 |
| DVV | Danville, Virginia | United States | PSU | -79.44 | 36.71 | 492.00 | 15.0 | C | 2016/07 | 2017/12 |
| EGB | Egbert, Ontario | Canada | EC | -79.78 | 44.23 | 276.00 | 25.0 | C | 2005/03 | 2020/12 |
| EIC | Easter Island | Chile | NOAA | -109.45 | -27.13 | 72.00 | 4.5 | D | 2000/01 | 2019/11 |
| ENA | Eastern North Atlantic, Graciosa, Azores | Portugal | LBNL-ARM | -28.03 | 39.09 | 40.48 | 25.0 | C | 2015/07 | 2019/12 |



| ESP | Estevan Point, British Columbia | Canada | EC | -126.54 | 49.38 | 47.00 | 25.0 | C | 2009/03 | 2020/12 |
|---|---|---|---|---|---|---|---|---|---|---|
| EST | Esther, Alberta | Canada | EC | -110.21 | 51.67 | 757.00 | 30.0 | C | 2010/01 | 2020/11 |
| ETL | East Trout Lake, Saskatchewan | Canada | EC | -104.99 | 54.35 | 598.00 | 30.0 | C | 2005/08 | 2020/10 |
| FNE | Fort Nelson, British Columbia | Canada | EC | -122.57 | 58.84 | 376.00 | 30.0 | C | 2014/07 | 2020/12 |
| FSD | Fraserdale | Canada | EC | -81.57 | 49.88 | 250.00 | 30.0 | C | 2000/01 | 2020/05 |
| GAT | Gartow | Germany | ICOS-ATC,HPB | 11.44 | 53.07 | 411.00 | 25.0 | C | 2016/05 | 2021/12 |
| GCI | Millerville, AL | United States | PSU | -85.89 | 33.18 | 428.00 | 25.0 | C | 2017/10 | 2018/05 |
| GMI | Mariana Islands | Guam | NOAA | 144.66 | 13.39 | 8.00 | 15.0 | D | 2000/01 | 2021/09 |
| GPA | Gunn Point | Australia | CSIRO | 131.04 | -12.25 | 37.00 | 75.0 | D | 2010/08 | 2021/02 |
| HBA | Halley Station, Antarctica | United Kingdom | NOAA | -26.21 | -75.61 | 35.00 | 4.5 | D | 2000/01 | 2021/02 |
| HEI | Heidelberg | Germany | IUP | 8.68 | 49.42 | 143.00 | 30.0 | C | 2005/01 | 2014/09 |
| HNP | Hanlan's Point, Ontario | Canada | EC | -79.39 | 43.61 | 97.00 | 25.0 | C | 2014/06 | 2020/12 |
| HPB | Hohenpeissenberg | Germany | ICOS-ATC,HPB | 11.02 | 47.80 | 1065.00 | 25.0 | C | 2015/09 | 2021/12 |
| HSU | Humboldt State University | United States | NOAA | -124.44 | 41.57 | 7.60 | 30.0 | D | 2008/05 | 2017/05 |
| HTM | Hyltemossa | Sweden | ICOS-ATC,LUND-CEC | 13.42 | 56.10 | 265.00 | 25.0 | C | 2017/04 | 2021/12 |
| ICE | Storhofdi, Vestmannaeyjar | Iceland | NOAA | -20.29 | 63.40 | 121.70 | 15.0 | D | 2000/01 | 2021/11 |
| IGR | Igrim | Russian Federation | NIES | 64.41 | 63.19 | 89.00 | 30.0 | C | 2005/04 | 2013/07 |
| INU | Inuvik,Northwest Territories | Canada | EC | -133.53 | 68.32 | 123.00 | 15.0 | C | 2012/02 | 2020/12 |
| IPR | Ispra | Italy | ICOS-ATC,JRC | 8.64 | 45.81 | 310.00 | 30.0 | C | 2007/10 | 2021/12 |
| IZO | Izana, Tenerife, Canary Islands | Spain | NOAA | -16.48 | 28.30 | 2377.90 | 25.0 | D | 2000/01 | 2021/10 |
| JFJ | Jungfraujoch | Switzerland | ICOS-ATC,HFSJG | 7.99 | 46.55 | 3585.00 | 15.0 | C | 2005/02 | 2021/12 |
| KEY | Key Biscayne, Florida | United States | NOAA | -80.20 | 25.67 | 6.00 | 25.0 | D | 2000/02 | 2021/12 |
| KIT | Karlsruhe | Germany | ICOS-ATC,HPB | 8.42 | 49.09 | 310.00 | 30.0 | C | 2016/12 | 2021/12 |
| KJN | Kjolnes | Norway | UEXE, MPI-BGC | 29.23 | 70.85 | 20.00 | 15.0 | C | 2013/10 | 2018/08 |
| KMP | Kumpula | Finland | FMI | 24.96 | 60.20 | 53.00 | 30.0 | C | 2010/01 | 2021/12 |
| KRE | Kresin u Pacova | Czech Republic | ICOS | 15.08 | 49.57 | 784.00 | 25.0 | C | 2017/04 | 2021/12 |
| KRS | Karasevoe | Russian Federation | NIES | 82.42 | 58.25 | 156.00 | 30.0 | C | 2004/09 | 2018/12 |
| KUM | Cape Kumukahi, Hawaii | United States | NOAA | -155.01 | 19.51 | 3.00 | 15.0 | D | 2000/01 | 2021/12 |
| LEF | Park Falls, Wisconsin | United States | NOAA | -90.27 | 45.95 | 868.00 | 30.0 | C | 2010/09 | 2021/12 |
| LEF | Park Falls, Wisconsin | United States | NOAA | -90.26 | 45.95 | 868.00 | 30.0 | D | 2000/01 | 2021/12 |
| LIN | Lindenberg | Germany | ICOS-ATC,HPB | 14.12 | 52.17 | 171.00 | 30.0 | C | 2015/10 | 2021/12 |
| LLB | Lac La Biche, Alberta | Canada | NOAA | -112.45 | 54.95 | 546.10 | 30.0 | D | 2008/01 | 2013/02 |
| LLB | Lac La Biche, Alberta | Canada | EC | -112.47 | 54.95 | 590.00 | 30.0 | C | 2007/04 | 2020/12 |
| LLN | Lulin | Taiwan | NOAA | 120.86 | 23.47 | 2867.00 | 25.0 | D | 2006/08 | 2021/11 |
| LMP | Lampedusa | Italy | ICOS-ATC,ENEA | 12.63 | 35.52 | 53.00 | 25.0 | C | 2020/01 | 2021/12 |
| LMT | Lamezia Terme | Italy | ISAC | 16.23 | 38.88 | 14.00 | 30.0 | C | 2015/01 | 2016/12 |
| LPO | Ile Grande | France | LSCE | -3.58 | 48.80 | 20.00 | 15.0 | D | 2005/01 | 2013/08 |
| LUT | Lutjewad | Netherlands | ICOS-ATC,RUG | 6.35 | 53.40 | 61.00 | 25.0 | C | 2006/05 | 2021/12 |
| MAA | Mawson, Antarctica | Australia | CSIRO | 62.87 | -67.62 | 32.00 | 4.5 | D | 2000/02 | 2021/02 |
| MEX | High Altitude Global Climate Observation Center | Mexico | NOAA | -97.31 | 18.98 | 4469.00 | 15.0 | D | 2009/01 | 2021/09 |
| MID | Sand Island, Midway | United States | NOAA | -177.38 | 28.21 | 8.00 | 15.0 | D | 2000/01 | 2021/11 |
| MKN | Mt. Kenya | Kenya | NOAA | 37.30 | -0.06 | 3649.00 | 25.0 | D | 2003/12 | 2011/06 |
| MLO | Mauna Loa, Hawaii | United States | NOAA | -155.58 | 19.54 | 3437.00 | 15.0 | C | 2000/01 | 2021/12 |
| MLO | Mauna Loa, Hawaii | United States | NOAA | -155.58 | 19.54 | 3402.00 | 15.0 | D | 2000/01 | 2021/12 |
| MNM | Minamitorishima | Japan | JMA | 153.98 | 24.29 | 27.10 | 15.0 | C | 2000/01 | 2020/12 |
| MQA | Macquarie Island | Australia | CSIRO | 158.97 | -54.48 | 13.00 | 4.5 | D | 2000/01 | 2021/02 |
| MRC | Marcellus Pennsylvania | United States | PSU | -76.42 | 41.47 | 652.00 | 75.0 | C | 2015/05 | 2018/12 |
| NAT | Farol De Mae Luiza Lighthouse | Brazil | NOAA | -35.19 | -5.51 | 20.00 | 15.0 | D | 2010/09 | 2020/03 |





| NGL | Neuglobsow | Germany | UBA | 13.03 | 53.14 | 62.00 | 75.0 | C | 2005/01 | 2013/12 |
|---|---|---|---|---|---|---|---|---|---|---|
| NMB | Gobabeb | Namibia | NOAA | 15.01 | -23.58 | 461.00 | 25.0 | D | 2000/01 | 2021/09 |
| NOR | Norunda | Sweden | ICOS-ATC,LUND-CEC | 17.48 | 60.09 | 146.00 | 15.0 | C | 2017/04 | 2021/12 |
| NOY | Noyabrsk | Russian Federation | NIES | 75.78 | 63.43 | 188.00 | 30.0 | C | 2005/10 | 2018/12 |
| NWR | Niwot Ridge, Colorado | United States | NOAA | -105.57 | 40.05 | 3526.00 | 15.0 | D | 2000/01 | 2021/11 |
| OPE | Observatoire perenne de l'environnement | France | ICOS-ATC,LSCE | 5.50 | 48.56 | 510.00 | 30.0 | C | 2011/07 | 2021/12 |
| OTA | Otway Basin | Australia | CSIRO | 142.82 | -38.52 | 50.00 | 30.0 | D | 2005/09 | 2014/08 |
| OXK | Ochsenkopf | Germany | ICOS-ATC,HPB | 11.81 | 50.03 | 1185.00 | 30.0 | C | 2006/06 | 2021/12 |
| PAL | Pallas-Sammaltunturi, GAW Station | Finland | ICOS-ATC,FMI | 24.12 | 67.97 | 577.00 | 15.0 | C | 2005/01 | 2021/12 |
| PDM | Pic du Midi | France | LSCE | 0.14 | 42.94 | 2887.00 | 15.0 | D | 2005/02 | 2018/02 |
| POC | Pacific Ocean | Pacific Ocean | NOAA | -130.75 | 0.12 | 20.00 | 15.0 | D | 2000/01 | 2017/07 |
| PSA | Palmer Station, Antarctica | United States | NOAA | -64.05 | -64.77 | 15.00 | 4.5 | D | 2000/01 | 2021/05 |
| PTA | Point Arena, California | United States | NOAA | -123.74 | 38.95 | 22.00 | 25.0 | D | 2000/01 | 2011/05 |
| PUI | Puijo | Finland | ICOS-ATC, UEF | 27.66 | 62.91 | 84.00 | 30.0 | C | 2011/11 | 2020/12 |
| PUY | Puy de Dome | France | ICOS-ATC,LSCE | 2.97 | 45.77 | 1475.00 | 15.0 | C | 2010/07 | 2021/12 |
| RPB | Ragged Point | Barbados | NOAA | -59.43 | 13.16 | 20.00 | 15.0 | D | 2000/01 | 2021/11 |
| RUN | La Réunion | France | ICOS-ATC,LSCE | 55.38 | -21.08 | 2160.00 | 15.0 | C | 2018/05 | 2021/12 |
| RYO | Ryori | Japan | JMA | 141.82 | 39.03 | 280.00 | 15.0 | C | 2000/01 | 2020/12 |
| SAC | Saclay | France | ICOS-ATC,CEA | 2.14 | 48.72 | 260.00 | 75.0 | C | 2017/05 | 2021/12 |
| SCT | Beech Island, South Carolina | United States | NOAA | -81.83 | 33.41 | 420.20 | 75.0 | C | 2015/08 | 2021/12 |
| SDZ | Shangdianzi | China | NOAA | 117.12 | 40.65 | 298.00 | 15.0 | D | 2009/09 | 2015/09 |
| SEY | Mahe Island | Seychelles | NOAA | 55.53 | -4.68 | 7.00 | 15.0 | D | 2000/01 | 2021/08 |
| SGP | Southern Great Plains, Oklahoma | United States | NOAA | -97.50 | 36.62 | 339.00 | 75.0 | D | 2002/04 | 2021/12 |
| SGP | Southern Great Plains, Oklahoma | United States | LBNL-ARM | -97.49 | 36.61 | 374.00 | 75.0 | C | 2011/01 | 2020/12 |
| SHM | Shemya Island, Alaska | United States | NOAA | 174.08 | 52.72 | 28.00 | 25.0 | D | 2000/01 | 2021/07 |
| SMO | Tutuila | American Samoa | NOAA | -170.56 | -14.23 | 60.30 | 15.0 | D | 2000/01 | 2021/11 |
| SMR | Hyytiala | Finland | ICOS-ATC,UHELS | 24.29 | 61.85 | 306.00 | 25.0 | C | 2016/12 | 2021/12 |
| SNB | Sonnblick | Austria | EAA | 47.05 | 12.96 | 3111.00 | 15.0 | C | 2012/04 | 2018/12 |
| SOD | Sodankylä | Finland | FMI | 26.64 | 67.36 | 227.00 | 25.0 | C | 2012/01 | 2021/12 |
| SPO | South Pole, Antarctica | United States | NOAA | -24.80 | -89.96 | 2821.30 | 4.5 | D | 2000/01 | 2021/10 |
| STE | Steinkimmen | Germany | ICOS-ATC,HPB | 8.46 | 53.04 | 281.00 | 75.0 | C | 2019/07 | 2021/12 |
| SUM | Summit | Greenland | NOAA | -38.42 | 72.60 | 3214.54 | 15.0 | D | 2000/08 | 2021/07 |
| SVB | Svartberget | Sweden | ICOS-ATC,SLU | 19.77 | 64.26 | 419.00 | 25.0 | C | 2017/06 | 2021/12 |
| SYO | Syowa Station, Antarctica | Japan | NOAA | 39.59 | -69.00 | 16.00 | 4.5 | D | 2000/01 | 2020/12 |
| TAC | Tacolneston | United Kingdom | NOAA | 1.14 | 52.52 | 236.00 | 25.0 | D | 2014/06 | 2016/01 |
| TAP | Tae-ahn Peninsula | Republic of Korea | NOAA | 126.13 | 36.73 | 21.00 | 75.0 | D | 2000/01 | 2021/10 |
| THD | Trinidad Head, California | United States | NOAA | -124.15 | 41.05 | 112.00 | 25.0 | D | 2002/04 | 2017/06 |
| TIK | Hydrometeorological Observatory of Tiksi | Russia | NOAA | 128.89 | 71.60 | 29.00 | 15.0 | D | 2011/08 | 2018/09 |
| TIK | Tiksi | Russian Federation | FMI | 128.89 | 71.60 | 29.00 | 15.0 | C | 2010/09 | 2020/12 |
| TOH | Torfhaus | Germany | ICOS-ATC,HPB | 10.53 | 51.81 | 948.00 | 25.0 | C | 2017/12 | 2021/12 |
| TPD | Turkey Point, Ontario | Canada | EC | -80.56 | 42.64 | 266.00 | 25.0 | C | 2012/11 | 2020/12 |
| TRN | Trainou | France | ICOS-ATC,LSCE | 2.11 | 47.96 | 311.00 | 25.0 | C | 2006/08 | 2021/12 |
| USH | Ushuaia | Argentina | NOAA | -68.31 | -54.85 | 32.00 | 4.5 | D | 2000/01 | 2021/06 |
| UTA | Wendover, Utah | United States | NOAA | -113.72 | 39.90 | 1332.00 | 25.0 | D | 2000/01 | 2021/12 |
| UTO | Uto | Finland | FMI | 21.37 | 59.78 | 65.00 | 25.0 | C | 2012/03 | 2018/03 |
| UTO | Uto | Finland | ICOS-ATC,FMI | 21.37 | 59.78 | 65.00 | 25.0 | C | 2018/03 | 2021/12 |
| UUM | Ulaan Uul | Mongolia | NOAA | 111.10 | 44.45 | 1012.00 | 25.0 | D | 2000/01 | 2020/10 |
| VGN | Vaganovo | Russian Federation | NIES | 62.32 | 54.50 | 277.00 | 30.0 | C | 2008/06 | 2018/12 |





| VKV | Voeikovo | Russian Federation | MGO | 30.70 | 59.95 | 76.00 | 25.0 | C | 2008/05 | 2014/12 |
| WIS | Weizmann Institute of Science at the Arava Ins... | Israel | NOAA | 35.06 | 29.96 | 482.00 | 25.0 | D | 2000/01 | 2021/11 |
| WLG | Mt. Waliguan | Peoples Republic of China | NOAA | 100.90 | 36.27 | 3890.00 | 15.0 | D | 2000/01 | 2021/09 |
| WPC | Western Pacific Cruise | Western Pacifi | NOAA | 143.70 | 0.13 | 10.00 | 15.0 | D | 2004/12 | 2013/06 |
| WSA | Sable Island, Nova Scotia | Canada | EC | -60.01 | 43.93 | 8.00 | 25.0 | C | 2003/06 | 2020/03 |
| YAK | Yakutsk | Russian Federation | NIES | 129.36 | 62.09 | 344.00 | 30.0 | C | 2007/09 | 2013/12 |
| YON | Yonagunijima | Japan | JMA | 123.01 | 24.47 | 50.00 | 30.0 | C | 2000/01 | 2020/12 |
| ZEP | Ny-Alesund, Svalbard | Norway and Sweden | ICOS-ATC,NILU | 11.89 | 78.91 | 489.00 | 15.0 | C | 2012/04 | 2021/12 |
| ZOT | Zotino | Russian Federation | MPI-BGC | 89.21 | 60.48 | 415.00 | 25.0 | C | 2009/05 | 2016/12 |
| ZOT | Zotino | Russian Federation | MPI-BGC | 89.21 | 60.48 | 415.00 | 15.0 | D | 2006/10 | 2013/06 |

*Sampling heights from which atmospheric $CH_4$ is sampled in TM5. **Observations used in this study between 2010 and 2021.



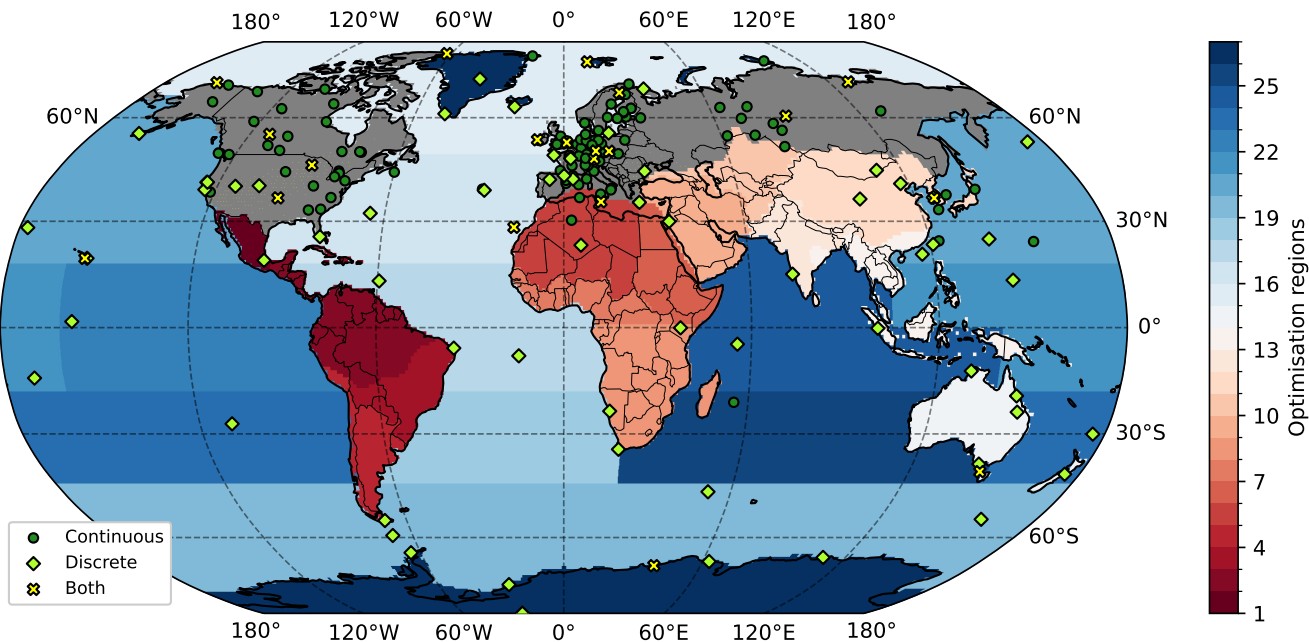

**Figure A1.** The locations of the atmospheric CH$_4$ measurement sites and the type of measurement used (continuous, discrete or both) in the inversions. The areas optimised regionally are shown with blues and reds, and the grey colour shows the area optimised at 1° latitude × 1° longitude resolution.



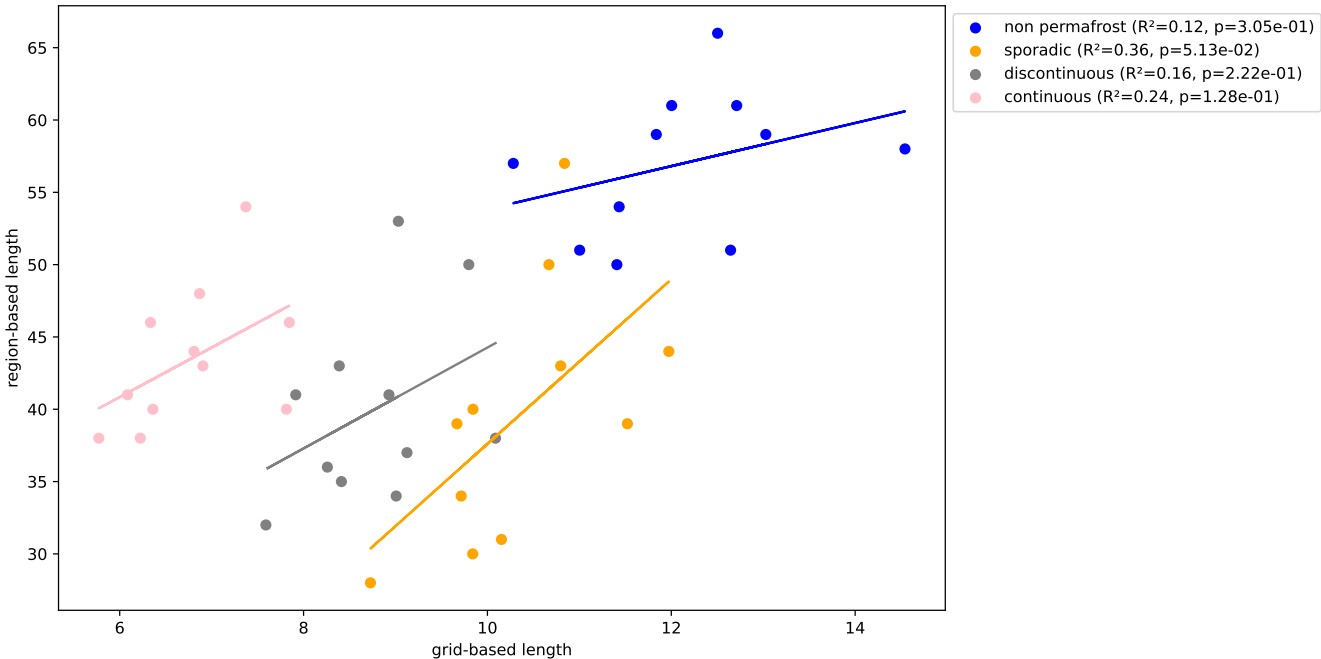

**Figure A2.** Relationship between grid-based and region-based melting seasons. $R^2$ and p in the legends are the coefficient of determination and p-values of the slopes from linear regression fit, indicating statistical significance of the coefficient of determination.

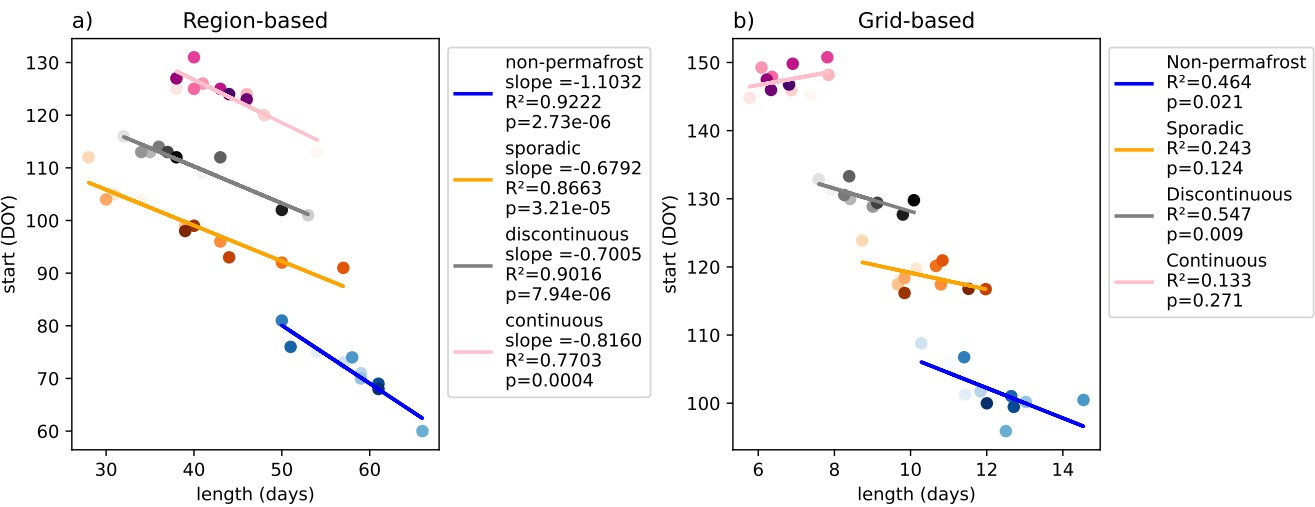

**Figure A3.** Relationship between region-based (a) and grid-based (b) melting seasons. $R^2$ and p in the legends are the coefficient of determination and p-values of the slopes from linear regression fit, indicating statistical significance of the coefficient of determination.



**Table A2.** Region-based melting season start and end days, as well as the length of the melting season in days for the four permafrost zones: non-permafrost, sporadic, discontinuous and continuous permafrost defined for each zone. The start and end days are represented as a number of days from the beginning of the year, with number one being the first day of the year (day-of-year).

|  | non-permafrost | | | sporadic | | | discontinuous | | | continuous permafrost | | |
|---|---|---|---|---|---|---|---|---|---|---|---|---|
|  | Start | End | Length | Start | End | Length | Start | End | Length | Start | End | Length |
| 2011 | 80 | 130 | 51 | 105 | 138 | 34 | 109 | 149 | 41 | 114 | 167 | 54 |
| 2012 | 76 | 129 | 54 | 106 | 136 | 31 | 110 | 150 | 41 | 126 | 163 | 38 |
| 2013 | 74 | 130 | 57 | 113 | 140 | 28 | 117 | 148 | 32 | 121 | 168 | 48 |
| 2014 | 72 | 130 | 59 | 100 | 138 | 39 | 102 | 154 | 53 | 125 | 170 | 46 |
| 2015 | 71 | 129 | 59 | 105 | 134 | 30 | 114 | 148 | 35 | 127 | 167 | 41 |
| 2016 | 61 | 126 | 66 | 97 | 139 | 43 | 114 | 147 | 34 | 126 | 165 | 40 |
| 2017 | 75 | 132 | 58 | 93 | 142 | 50 | 115 | 150 | 36 | 132 | 171 | 40 |
| 2018 | 82 | 131 | 50 | 92 | 148 | 57 | 113 | 155 | 43 | 126 | 168 | 43 |
| 2019 | 77 | 127 | 51 | 94 | 137 | 44 | 114 | 150 | 37 | 128 | 165 | 38 |
| 2020 | 70 | 130 | 61 | 100 | 139 | 40 | 103 | 152 | 50 | 124 | 169 | 46 |
| 2021 | 69 | 129 | 61 | 99 | 137 | 39 | 113 | 150 | 38 | 125 | 168 | 44 |
| mean | 73 | 129 | 57 | 100 | 139 | 40 | 111 | 150 | 40 | 125 | 167 | 43 |

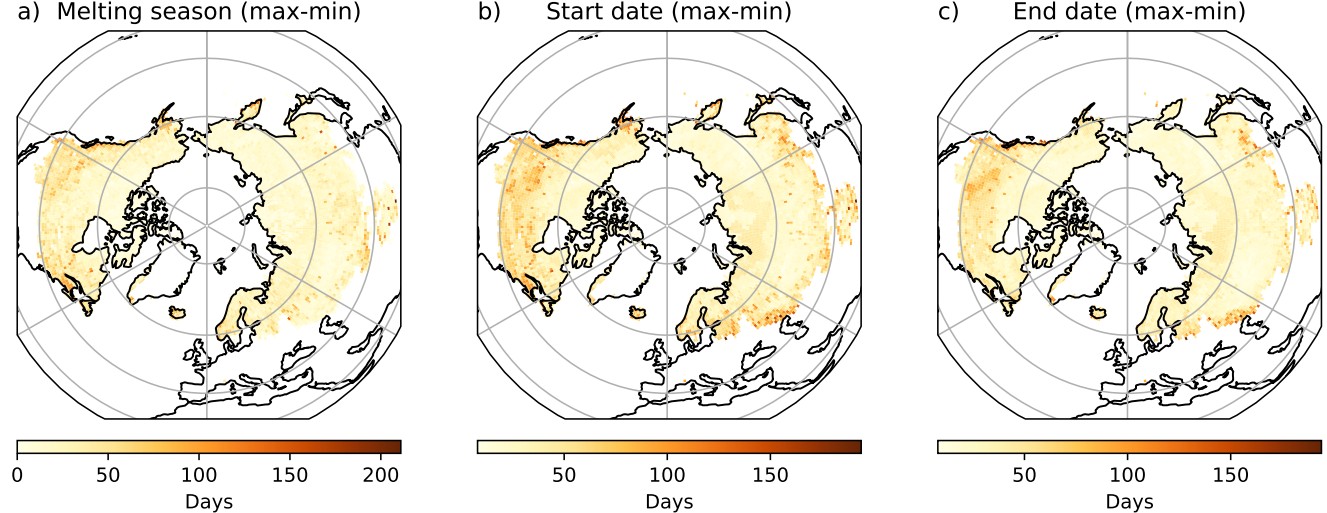

**Figure A4.** Grid-based difference between the maximum and minimum length (a), start day (b) and end day (c) of the melting season during the study period (2011-2021).



**Table A3.** Region-based melting season methane emissions in the four permafrost zones; non-permafrost, sporadic, discontinuous, and continuous permafrost, as well as the total sum of the emissions.

|  | non-permafrost (Gg) | sporadic (Gg) | discontinuous (Gg) | continuous permafrost (Gg) | total emissions (Gg) |
|---|---|---|---|---|---|
| 2011 | 1293.7 | 205.1 | 116.9 | 315.5 | 1931.1 |
| 2012 | 1014.6 | 206.2 | 127.6 | 209.3 | 1557.7 |
| 2013 | 1121.9 | 209.3 | 99.5 | 277.0 | 1707.7 |
| 2014 | 1248.4 | 294.4 | 183.4 | 327.9 | 2054.1 |
| 2015 | 1275.5 | 210.3 | 150.6 | 249.1 | 1885.6 |
| 2016 | 1378.8 | 271.8 | 132.2 | 201.2 | 1984.0 |
| 2017 | 1268.7 | 266.2 | 121.8 | 254.3 | 1911.1 |
| 2018 | 1102.7 | 302.4 | 162.8 | 261.1 | 1829.0 |
| 2019 | 1063.5 | 231.5 | 124.5 | 161.1 | 1580.5 |
| 2020 | 1216.6 | 250.3 | 144.6 | 253.7 | 1865.2 |
| 2021 | 1195.7 | 230.2 | 129.5 | 271.9 | 1827.2 |
| mean | 1198.2 | 243.4 | 135.8 | 252.9 | 1830.3 |

**Table A4.** Grid-based melting season emissions in the four permafrost zones; non-permafrost, sporadic, discontinuous, and continuous permafrost, as well as the total sum of the emissions.

|  | non-permafrost (Gg) | sporadic (Gg) | discontinuous (Gg) | continuous permafrost (Gg) | total emissions (Gg) |
|---|---|---|---|---|---|
| 2011 | 257.2 | 63.2 | 36.9 | 52.3 | 409.5 |
| 2012 | 260.8 | 77.2 | 27.0 | 32.2 | 397.2 |
| 2013 | 257.1 | 62.3 | 21.5 | 36.6 | 377.5 |
| 2014 | 304.1 | 84.8 | 37.0 | 49.6 | 475.5 |
| 2015 | 333.2 | 70.6 | 34.8 | 35.9 | 474.5 |
| 2016 | 356.3 | 75.9 | 37.0 | 38.6 | 507.8 |
| 2017 | 402.5 | 75.8 | 32.3 | 45.7 | 556.3 |
| 2018 | 310.0 | 81.0 | 32.3 | 42.9 | 466.1 |
| 2019 | 350.2 | 77.0 | 34.6 | 28.0 | 489.8 |
| 2020 | 281.7 | 64.7 | 35.2 | 27.5 | 409.1 |
| 2021 | 255.3 | 74.5 | 36.9 | 35.4 | 402.0 |
| mean | 306.2 | 73.4 | 33.2 | 38.6 | 451.4 |





**Figure A5.** Grid-based melting season methane emissions and the mean temperature in the four permafrost zones; a) non-permafrost, b) sporadic, c) discontinuous, and d) continuous permafrost. Grid-cells with negative emission rate were masked out. $R^2$ and p in the legends are the coefficient of determination and p-values of the slopes from linear regression fit, indicating statistical significance of the coefficient of determination.





**Table A5.** Melting season methane emissions in the Hudson Bay lowlands and Western Siberian lowlands.

|  | region-based [Gg] | | | grid-based [Gg] | | |
|---|---|---|---|---|---|---|
|  | HBL | WSL | all | HBL | WSL | all |
| 2011 | 84.4 | 514.0 | 598.4 | 24.7 | 73.3 | 97.9 |
| 2012 | 115.7 | 393.6 | 509.2 | 44.3 | 81.7 | 125.9 |
| 2013 | 91.9 | 407.6 | 499.5 | 21.0 | 89.1 | 110.1 |
| 2014 | 116.9 | 533.1 | 650.0 | 41.2 | 137.8 | 179.0 |
| 2015 | 70.7 | 609.8 | 680.5 | 17.4 | 142.4 | 159.8 |
| 2016 | 81.6 | 514.6 | 596.3 | 22.4 | 126.2 | 148.5 |
| 2017 | 101.6 | 532.6 | 634.2 | 30.4 | 123.2 | 153.5 |
| 2018 | 111.6 | 420.6 | 532.2 | 39.4 | 113.4 | 152.9 |
| 2019 | 84.5 | 334.2 | 418.7 | 28.8 | 119.3 | 148.1 |
| 2020 | 114.2 | 461.5 | 575.7 | 45.7 | 89.7 | 135.4 |
| 2021 | 98.0 | 446.3 | 544.3 | 29.1 | 108.7 | 137.8 |
| mean | 97.4 | 469.8 | 567.2 | 31.3 | 109.5 | 140.8 |



**Figure A6.** Region-based emission maps of the melting season in the years that had some of the highest emissions in the Hudson Bay lowlands and Western Siberian lowlands. Hudson Bay lowlands have been outlined with red borders, and Western Siberian lowlands with blue borders.





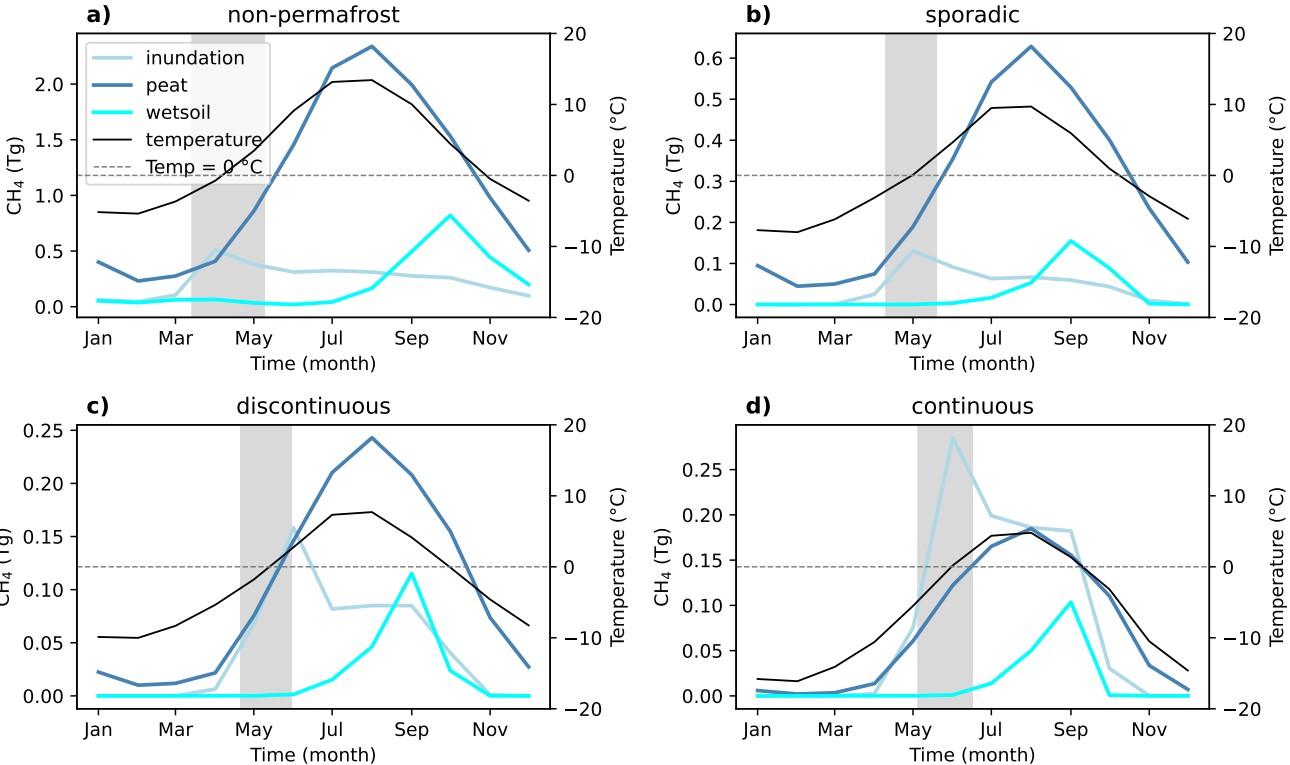

**Figure A7.** Average monthly emissions from peat and inundation for the years 2011-2019, from the prior LPX–Bern DYPTOP v1.4 emissions. The average region-based melting season is showed with gray shading.



*Author contributions.* S.H., M.T. and T.A. participated in the design of the study. T.A. supervised the project. M.T and T.A. offered advice of the analysis of the results. M.T. performed the model runs with the CTE-CH$_4$ and A.T. helped in setting up the model runs and interpretation of the results. S.H. did the data-analysis and prepared the visualizations, as well as wrote the original manuscript with the help of M.T. and T.A. K.R. provided and helped to interpret the SMOS F/T soil state data. A.E. postprocessed the SMOS F/T data for the analysis and helped to interpret the results of the SMOS F/T data. M.S. provided CH$_4$ mole fraction measurements from sites in Western Siberian lowlands. H.A. provided CH$_4$ mole fraction measurements from Kumpula and Sodankylä sites. All authors have read and commented and approved the published version of the manuscript.

*Competing interests.* The authors declare no conflict of interest. Also, the funders had no role in the design of the study; in the collection, analysis, or interpretation of data; in the writing of the manuscript, or in the decision to publish the results.



*Acknowledgements.* We thank the team behind the LPX-Bern DYPTOP v1.4 for providing the $CH_4$ emission estimates. The authors would like to thank the ICOS and ICOS-Finland PIs for providing the data on $CH_4$ mole fractions. We thank the Finnish Meteorological Institute (PAL, UTO, KMP, SOD), University of Eastern Finland (PUI) and University of Helsinki (SMR) for providing the methane data in Finland. We are grateful for CSIRO Oceans and Atmosphere, Climate Science Centre (CSIRO), Environment and Climate Change Canada (ECCC), the Hungarian Meteorological Service (HMS), the Institute for Atmospheric Sciences and Climate (ISAC), the Institute on Atmospheric Pollution of the National Research Council (IIA), the Institute of Environmental Physics, University of Heidelberg (IUP), Laboratoire des Sciences du Climat et de l'Environnement (LSCE), Lawrence Berkeley National Laboratory (LBNL-ARM), the Environment Division Global Environment and Marine Department Japan Meteorological Agency (JMA), the Main Geophysical Observatory (MGO), the Max Planck Institute for Biogeochemistry (MPIBGC), National Institute for Environmental Studies (NIES), Norwegian Institute for Air Research (NILU), National Oceanic and Atmospheric Administration Earth System Research Laboratories (NOAA ESRL), the Pennsylvania State University (PSU), Swedish University of Agricultural Sciences (SLU), the Swiss Federal Laboratories for Materials Science and Technology (EMPA), Umweltbundesamt Germany/Federal Environmental Agency (UBA), Umweltbundesamt Austria/Environment Agency Austria (EAA) as the data provider for Sonnblick, University of Bristol (UNIVBRIS), University of Exeter (Univ. Exeter), and University of Urbino (UNIURB) for performing high-quality $CH_4$ measurements at global sites and making them available through the Global Atmosphere Watch - World Data Centre for Greenhouse Gases (GAW-WDCGG) and personal communications.

*Financial support.* We thank the European Space Agency ESRIN Contract No: 4000124500/18/I–EF (SMOS F/T Service) 2:44, 4000125046/18/I–NB (MethEO), 4000137895/22/I–AG MethaneCAMP, AO/1–10901/21/I–DT AMPAC–Net, FIRI – ICOS Finland (345531), ICOS–ERIC (281250), and 351311 (GHGSUPER), 337552 and 359196 (Flagships ACCC and FAME), ESA AO/1-11844/23/I-NS SMART–CH4 and the Research Council of Finland (364034) for financial support.



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
