# Peer review of "Spring melting season methane emissions in northern high latitude wetlands are governed by the length of the season and presence of permafrost"

_EGUsphere, 2025_

## Referee Comment (RC1)

The authors present a study on estimating methane emissions in northern high latitudes during the spring melting season using two different inverse modeling set-ups. The study also includes a detailed description and analysis of the estimation of the spring melting season in four different permafrost zones using SMOS F/T data. This study is a valuable contribution in reducing uncertainties in estimating the length of the spring melting season in permafrost regions, which are increasingly affected by climate change, as well as estimating the resulting CH4 emissions in those regions.

In my opinion, this study is well prepared and carefully thought out, so only minor improvements are necessary. Specifically, I believe there are three points that should be taken into account when revising the manuscript:

1. There are a few details about the inverse modeling set-up and the analysis that I didn't quite understand from the description. You describe the model domain of the transport model, but are the CH4 fluxes also optimized over the whole domain (globally)? And in the results, do you evaluate the total CH4 fluxes (anthropogenic+natural) or only the natural fluxes or only the biogenic fluxes? If you evaluate all natural fluxes, what about the contribution from other natural sources, since you predominantly want to estimate wetland emission? Please add these details in your description.

2. In some cases, the figures shown are not adequately described in the accompanying text. For example, Fig. 4 has four sub-figures and displays multiple set-ups, but there is only one brief sentence describing it. Please ensure that all figures are described properly and that it is clear what they display.

3. Since you are using inverse modeling, it would also be useful to include a brief comparison with the prior fluxes. So far, this has only been mentioned briefly in the discussion. For instance, you state that the CH4 fluxes are highest in the non-permafrost zone, which you attribute to it being the largest zone. However, could this also be related to the fact that the prior was possibly already estimated to be higher in these areas? The inversion can only optimize the fluxes to a certain extent. For example, if there are large areas in the continuous permafrost zone where the prior fluxes are zero, these will remain zero in the posterior state. These are some points to consider when discussing the results.

**Specific comments**

P1, L20-L22:
Could you specify "a large portion of the total soil carbon" with numbers?

P1, L20- P2, L30:
I think the link between permafrost thaw, the carbon stock and CH4 emissions needs to be emphasized more. From the section it is not clear, how the increased near-term CH4 emissions that are concluded at the end come about.

P2, L36-L38:

Could you give a source for the changing hydrology? Or is that sill de Vrese et al. (2023)?

P3, L68 :
"Another type of modeling is inverse modeling" Better: "Another approach to estimate fluxes is inverse modeling."

P3, Section 2.1:
Maybe it would be useful to give a short definition of "L-band", since the term is used repeatedly and is quite specific.

P4, L100-L101:
"Of the three categories, the thawing state of the soil is used in this study" but which one is the thawing state? The "partially frozen" or "thawed soil"? Maybe change the wording to clarify.

P4, Section 2.2:
Also put references to sections 2.2.2 and 2.2.3 to clarify, where you describe the corresponding observations and fluxes

P5, Section 2.2.3:
I read in the discussion that also the sink from the soil was included in the biospheric fluxes? This should already be mentioned here for clarification.

P6, L165:
"Areas where no SMOS F/T data was available, were excluded from this study". Could you roughly estimate the proportion of excluded areas?

P7, L186-L187:
"The boundaries used in this study were similar to the ones used by Erkkilä et al. (2023) to define different seasons in the northern high latitude wetlands." It would be good if you could still briefly indicate the boundaries in this paper, because "similar to", is too vague.

P7, Section 2.5.1:
In this section, you mention the word "spring" several times, e.g. L194 "the last day of spring". I'd be interested if you define this spring still based on e.g. month or exclusively by the melting season?

P9, L243-L244:
These average lengths values represent the average over all permafrost zones? If so, it should be added in the sentence for clarity.

P10, L198-L291:
"The mean values of the length and temperature of the grid-based melting season might not have been the best to describe the relationship..." Since you state that it's not the best way, what would be a better method in your opinion?

P12, Fig. 3:
Could you add more spaces between the tick labels in the lower panels? It would be easier to read.

P12, L317 – P13, L318:
"Hudson Bay lowlands and Western Siberian lowlands are some of the largest methane emitting wetlands in the northern high latitudes." Do you have a source for that or was it concluded from your emission dataset?

P14, L335:
"The average annual region-based melting season emissions" Does that include anthropogenic emissions or only natural?

P15, L371:
"Figures 7 and A6" Is there a reason why one of the figures is in the main text and the other in the supplements?

P23, L506:
"as the whole area is not permafrost or wetlands" Could you please re-formulate to clarify this wording?

P24, L538-L548:
Are the high share of 31% share of the emissions in the Hudson Bay and Western Siberian lowlands also related to higher emission estimates in the prior fluxes? Also, did you adjust the scale of the NIES measurements before including them in the inverse modeling framework?

**Technical corrections**

P1, L5:
Missing article? "for three permafrost zones and for **a** seasonally frozen non-permafrost zone"

P1, L17 to L20:
Consider splitting the sentence in two sentences. Also "**over a** 100-year timescale"

P2, L34:
"from **the** increasingly dry Arctic"

P2, L57:
No article: "but reliable soil temperature data"

P3, L60:
"at **a** resolution of 25 km"

P3, L62:
Singular: "for the whole northern high latitude region"

P3, L66:
"spring $CH_4$ emissions **have** been studied"

P3, L71:

"spring melting season and its CH4 emissions" Better: "spring melting season and the corresponding CH4 emissions"

P3: L76:
"**the** spring melting season"

P4, L104:
"the melting snow during daytime affect**s** the descending orbit"

P6, L160:
One set of parentheses too much: "(Obu et al., 2021)"

P6, L164:
"data had **no** values" ?

P6: L167-L168:
Please check sentence structure, it's not clear

P8, L208:
"There were **a** maximum of 18 SMOS F/T pixels and **a** minimum of"

P9, L255:
"Other areas with **a** noticeably longer melting season"

P10, L183:
Missing comma: "Additionally**,** there was a positive correlation"

P10, L286:
"not as strong **a** correlation"

P15, L343:
Fig. 8 is mentioned before Fig 7, they should probably be switched?

P16, L378:
"However, there is **no** major difference"

P22, L465:
One comma too much "Even though methane emissions"

P24, L530 – L531
One "model" too much "with multiple process-based ecosystem models for the northern wetlands"

P24, L534:
"the mean annual emissions in **the** Hudson Bay lowlands and Western Siberian lowlands"

P25, L556:
No dashes "melting season lengths varied more from year to year than."

P25, L562:
"in estimating **the** spring melting season"